# GRPO-λ : Credit Assignment improves LLM Reasoning

## Abstract

Large language models (LLMs) are increasingly deployed for tasks requiring complex reasoning, prompting significant interest in improving their reasoning abilities through post-training. Especially RL based methods using verifiable reward, like the state-of-the-art GRPO, have shown to tremendously improve reasoning behaviors when applied as post-training methods. However, the lack of an explicit reward or critic model limits GRPO's ability to assign fine-grained credit across token sequences. In this work, we present GRPO-λ , a novel extension to GRPO that enhances credit assignment in RL finetuning of LLMs for complex reasoning tasks. We approximate learning from λ-return with a reformulation of eligibility traces using token-level log-probabilities applied after each sequence generation, and a novel critic-free approximation of the temporal-difference error. We introduce a few variations for the weighting of the λ-return, and their applications to the *eligibility*-trace, where all the variations provide significant gains over GRPO. We compare GRPO-λ against GRPO by training models from 1.5B to 7B parameters on 4 different math reasoning datasets. The training plots demonstrate 30-40% improved performance during RL training on both LLaMA-3.1 and Qwen-2.5 architectures. Finally, we show that with GRPO-λ , the resulting average performance on AIME24, Math500, OlympiadMath, MinervaMath, and AMC improves over GRPO by over 3 points and a 4.5 points improvement on the 7B model.

## 1 Introduction

There is now a widespread acceptance of large language models (LLMs), wherein they are consulted on problems ranging from mundane tasks to ones requiring involved reasoning. For the latter, classical pre-training has been deemed insufficient due to the lack of explicit reasoning elicitations in the training data (Rajani et al., 2019). Thus, the focus to improving the reasoning skills of LLMs has been to expose them to problems requiring logic, such as mathematics and coding tasks, instead of aiming to produce plausible and coherent text (Hui et al., 2024; Xu et al., 2024; Yang et al., 2024; Shao et al., 2024). The recipe for scaling the performance on these reasoning tasks rests on elaborate post-training methods, including techniques like supervised-finetuning (SFT, Luo et al. 2023), reinforcement learning (RL, Schulman et al. 2017) without or with human feedback (RLHF, Ouyang et al. 2022), hybrids such as direct preference optimization (DPO, Rafailov et al. 2023), or any of their combinations.

Among these post-training techniques, RL shows promise as it transforms the next-token prediction problem to a reward maximization problem, allowing the LLM to freely generate new tokens as long as the resulting sequence produces satisfactory rewards. This is particularly relevant for reasoning problems such as mathematics and coding tasks, as the LLM needs to learn strategies that produce a verifiable, ground-truth outcome (e.g., the solution of the math problem). Recently, Deepseek-R1 (Guo et al., 2025) proposed an RL-based post-training method that resulted in the famously known "Aha! moment", where the model learned to perform self-reflection strategies. At its core lies group relative policy optimization (GRPO, Shao et al. 2024), which updates the LLM parameters using Monte-Carlo estimates of the policy returns to reinforce positive reasoning.

Contrary to the widely used PPO algorithm, GRPO does not require a critic to estimate the expected return of the policy. Instead, the expected return is approximated by taking the average over multiple rollouts of the policy. This makes GRPO lightweight, as there is no additional memory footprint for

the critic. However, what it does not do, contrary to PPO, is to use *eligibility traces* to update not only the current token based on the next one, but earlier tokens as well.

In RL, eligibility traces are a way to combine Monte-Carlo (MC) estimates and Temporal-Difference (TD) updates. It allows for rapid backpropagation of values to earlier states, and improves learning stability, as it balances between the high bias resulting from TD updates and the high variance resulting from MC estimates. This balance is governed by a parameter $\lambda \in [0, 1]$, where $\lambda = 0$ results in a pure TD update, and $\lambda = 1$ only uses the MC estimates. Importantly, this interpolation between one-step TD and MC methods is used to update PPO.

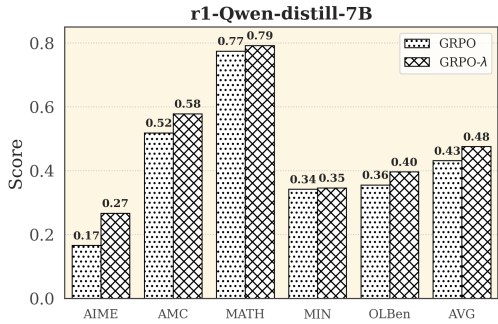

Figure 1: GRPO-$\lambda$ improves R1-Distill-Qwen-7B's evaluation performance by over 4 points on average across math benchmarks including a 10 point improvement on AIME24.

In this work, we reformulate these traces such that they are directly applicable to the policy. This allows us to combine eligibility traces with the value estimates from the rollouts produced by GRPO. We thus keep the advantages of GRPO, namely the lightweight memory footprint, while drastically improving the credit assignment through rapid value propagation towards earlier tokens. Moreover, in the setting of LLM post-training, reformulating eligibility traces for the policy can be seen as a form of token-specific weighting of the policy gradient loss. This insight leads us to propose different token-specific weighting mechanisms for credit assignment. Finally, thinking about eligibility traces made us focus on GRPO's value estimates at a token level. Since all GRPO's rollouts are performed from the same start-state (i.e., the prompt with the question to solve), its value estimates become increasingly inaccurate for later tokens in the sequence. We bound this error, which may be of independent interest to the reader.

To summarize, the contributions are:

1. With Lemma 1 and Theorem 1 we propose, GRPO-$\lambda$ , that extends GRPO through credit assignment with a novel reparameterization for PPO's eligibility trace, in a critic-free TD learning for language reasoning.

2. Finetuning different sized models and architectures on 4 different mathematical reasoning tasks show that GRPO-$\lambda$ learns faster and improves 30-40% better than GRPO on mathematical benchmarks tasks (Figure 4).

3. Benchmark performance of GRPO-$\lambda$ on 5 benchmarks shows an average increase of 3 points over GRPO (Table 1). And, for Deepseek-R1-Distill-Qwen-7B GRPO-$\lambda$ improves over 4 points (Figure 1).

4. Using insights from the proposed bounds to explore alternate trace weight styles, showing that for RL post-training of LLMs there are viable alternatives to the classic traces (Figure 3, Appendix D).

## 2 BACKGROUND

**Related work**  It has been an active field of research to distill deliberate reasoning abilities into LLMs, as they are often prone to quick judgments (Li et al., 2025). Early approaches attempted to explicitly instill reasoning into language models via explanations in training data, an expensive avenue as it requires large amounts of human-annotated data (Rajani et al., 2019; Nye et al., 2022). Chain-of-Thought (CoT) prompting provides a training-free alternative by simply prompting the model to think step-by-step (Wei et al., 2022b; Kojima et al., 2022), with potentially self-verification steps (Li et al., 2023; Wei et al., 2022b) or diversification of reasoning paths (Wang et al., 2023b; Fu et al., 2023). A logical next step has been to use self-generated CoT as a training signal for LLMs to iteratively improve their reasoning abilities (Zelikman et al., 2022). This is often done using RL (Trung et al., 2024). While the reward is usually provided at the end of the sequence (Singh

et al., 2024), as is the case for our setting, other works have tried to improve the credit assignment of intermediate steps using tree-search, at the expense of additional computations (Feng et al., 2023; Zhang et al., 2024). Finally, the provided reward is of crucial importance for the learned reasoning to be generalizable (Yeo et al., 2025). We refer to a broader overview of related work in Appendix F.

**Reinforcement Learning**  RL aims to solve a sequential decision problem, which can be modeled as a Markov Decision Problem (MDP) (Puterman, 1994) $(\mathcal{S}, \mathcal{A}, \mathcal{P}, \mathcal{R}, \gamma)$. $\mathcal{S}$ is the set of all possible states the environment can be in. $\mathcal{A}$ is the set of all possible actions that are available to the agent. $\mathcal{P} : \mathcal{S}, \mathcal{A} \to \mathcal{S}$ encompasses the environment's (stochastic) transition dynamics, $\mathcal{R} : \mathcal{S}, \mathcal{A} \to \mathbb{R}$ is the reward function and $\gamma$ is the discount factor. The agent can interact with the environment through a policy $\pi : \mathcal{S}, \mathcal{A} \to [0, 1]$ which maps a state to a probability distribution over the action-space conditioned on the state. At each timestep $t$, the agent receives the current state as input $s_t \in \mathcal{S}$ and takes the action $a_t \sim \pi(\cdot|s_t)$. The environment state is updated following the transition function $s_{t+1} \sim \mathcal{P}(\cdot|s_t, a_t)$ and gives a feedback to the agent in the form of a reward $r_t = \mathcal{R}(s_t, a_t)$.

We define the episodic return $G_t$ as the summation of the discounted rewards obtained by an agent along a trajectory following a policy $\pi$ and starting from timestep $t$. $G_t = \sum_{k=t}^{T} \gamma^{k-t} r_k$, where $T$ denotes the timestep at which the episode terminates. We further define the value function $V_\pi(s) = \mathbb{E}_\pi[G_t|s_t = s]$ which evaluate the expected episodic return of an agent following policy $\pi$ and starting at a specific state $s_t$. The goal of RL is to find the optimal policy $\pi^* = \mathrm{argmax}_\pi\{V_\pi(s_0)\}$ where $s_0$ follows the initial state distribution of the environment.

In the context of LLM post-training, the MDP definition is peculiar: $\mathcal{A}$ represents all possible tokens that can be generated by the LLM, and the state $s_t$ consists of a sequence of generated tokens, $s_t = (s_{t-1}, a_{t-1})$. For mathematical problems, the start-state $s_0$ consists of a mathematical question (also called prompt), tokenized to $m$ tokens, i.e., $s_0 = (a^0, \ldots, a^{m-1})$. The policy, in this case the pretrained LLM, selects the next token $a_t$ based on all previous tokens $s_t$. A special end-of-sequence (EOS) action $a^{\mathrm{EOS}}$ indicates the end of an episode. At that point, the generated answer is verified for correctness, resulting in $r_T = 1$ for correct answers, and $r_T = 0$ otherwise. All intermediate rewards are 0. This means that $G_0 = \gamma^T r_T \in [0, 1]$.

**PPO**  The fact that a pretrained LLM can be used as a good initial policy makes actor-critic methods, that explicitly represent a policy, such as PPO (Schulman et al., 2017), a particularly good fit for this setting. PPO is composed of an actor, the policy $\pi_\theta$ parametrized by $\theta$, and of a critic $V_\psi$ parametrized by $\psi$, which is used to estimate the expected return.

The use of $V_\psi$ provides a major benefit. With it, there is no need to wait until the episode ends to estimate $G_t$. Instead, one can bootstrap $G_t$ using $V_\psi$, e.g., $\hat{G}_t = r_t + \gamma r_{t+1} + \cdots + \gamma^{n-1} r_{t+n-1} + \gamma^n V_\psi(s_{t+n})$. With $n = T$, this falls back to the episodic return $G_t$, resulting in potentially high variance in returns between episodes due to the stochasticity of $\pi_\theta$. With $n = 1$, we mitigate the variance issue, but this introduces bias if $V_\psi$ is inaccurate. The difference between the 1-step $\hat{G}_t$ and the predicted value is also called the temporal-difference (TD) error $\delta_t = r_t + \gamma V_\psi(s_{t+1}) - V_\psi(s_t)$. A way to nicely balance this variance-bias trade-off is through generalized advantage estimation (GAE), which computes a weighted sum over TD errors, $A_{\mathrm{GAE}}(s_t) = \delta_t + \gamma\lambda\delta_{t+1}$, with $\lambda \in [0, 1]$ the weighting coefficient, and can be also seen as a weighted trace over future TD errors.

PPO combines GAE with a clipped surrogate objective function to update its policy, $\ell_{\mathrm{GAE}} = \min\left(\pi_{\mathrm{ratio}}(s_t)A_{\mathrm{GAE}}(s_t), \mathrm{clip}(\pi_{\mathrm{ratio}}(s_t), 1 - \epsilon, 1 + \epsilon)A_{\mathrm{GAE}}(s_t)\right)$, where $\pi_{\mathrm{ratio}}(s_t) = \frac{\pi_\theta(a_t|s_t)}{\pi_{\theta_{\mathrm{old}}}(a_t|s_t)}$ is the ratio between the current policy $\pi_\theta$ and the policy at the start of the epoch $\pi_{\theta_{\mathrm{old}}}$, and clipping $\pi_{\mathrm{ratio}}$ between $1 - \epsilon$ and $1 + \epsilon$ discourages $\pi_\theta$ from changing too much from $\pi_{\theta_{\mathrm{old}}}$ which, combined with GAE stabilizes learning. To update its critic $V_\psi$, PPO minimizes a mean-squared error (MSE) loss on the return, which in this case is bootstrapped using GAE, i.e., $\ell_\psi = \mathrm{MSE}\left(V_\psi(s_t), \mathrm{sg}(V_\psi(s_t) + A_{\mathrm{GAE}}(s_t))\right)$, where $\mathrm{sg}(.)$ is the stop-gradient operator. Additionally, specifically for LLM post-training, to avoid reward hacking (Trung et al., 2024; Yeo et al., 2025), PPO is combined with a KL-divergence regularizer on the initial, pretrained policy (also called the *referent policy*) $\pi_{\mathrm{ref}} := \pi_{\theta_0}$, i.e., $\ell_{\mathrm{KL}} = \mathrm{D}_{\mathrm{KL}}(\pi_\theta||\pi_{\mathrm{ref}})$. Combined, this results in the following PPO objective: $\ell_{\mathrm{PPO}} = \ell_\psi + \ell_{\mathrm{GAE}} - \beta\ell_{\mathrm{KL}}$, where $\beta$ is a small constant factor to weight the regularizer term.

**GRPO** All the benefits of PPO's critic $V_\psi$ rely on the fact that $V_\psi$ is decently accurate. In practice, for LLM post-training, this is a non-trivial task. First, the reward is sparse, only providing a binary signal at the end of each sequence generation. This complicates the task of $V_\psi$, which should be accurate at every intermediate token. Second, $\pi_\theta$, having been pretrained before starting RL post-training, is already far better than a random policy. This contrasts with $V_\psi$, who is often initialized to from a reward model (Huang et al., 2024), instead of predicting the policy's expected return. This disparity between $\pi_\theta$ and $V_\psi$ means $V_\psi$ has to "catch up" to $\pi_\theta$, which can hamper post-training. Next to the challenges of training $V_\psi$, it is also memory intensive, as $V_\psi$ has to be kept in memory with $\pi_\theta$. GRPO (Shao et al., 2024), a recent extension of PPO, aims to tackle these challenges by removing $V_\psi$ altogether. Instead, for a given prompt (i.e., a given start-state $s_0$), GRPO generates *multiple responses*, called a group $\mathcal{G} = \left\{ s_{0:T}^0, \ldots, s_{0:T}^{g-1} \right\}$, where $g = |\mathcal{G}|$ is a hyper-parameter denoting the size of the group. The group's average return is then used to approximate $V(s_0)$. Note that, since there is no critic, GRPO does not use GAE. Instead, the advantage is computed using a normalized advantage estimation (NAE), i.e., $A_{\text{NAE}}(s_t^i) = \frac{G_t^i - \mu_t^i}{\sigma_t^i}$ with $i \in [g]$, where $\mu_t^i, \sigma_t^i$ are the mean, standard deviation of all states $\left\{ s_t^0, \ldots, s_t^{g-1} \right\}$ in group $\mathcal{G}$. $A_{\text{NAE}}$ then replaces $A_{\text{GAE}}$ in PPO's surrogate objective function, i.e., $\ell_{\text{NAE}} = \min\left( \pi_{\text{ratio}}(s_t^i) A_{\text{NAE}}(s_t^i), \text{clip}(\pi_{\text{ratio}}(s_t^i), 1 - \epsilon, 1 + \epsilon) A_{\text{NAE}}(s_t^i) \right)$. This results in the following GRPO objective: $\ell_{\text{GRPO}} = \ell_{\text{NAE}} - \beta \ell_{\text{KL}}$.

## 3 GRPO-$\lambda$ FOR RAPID REWARD PROPAGATION

GRPO provides an efficient alternative to the PPO critic, avoiding its additional memory requirements and approximating the expected return with multiple Monte-Carlo rollouts. The use of $A_{\text{NAE}}$, however, comes with two downsides. First, since all the sequence generations from the same group were performed from the same state $s_0$, the baseline $\mu_t^i$ only estimates the expected return when $t = 0$, and is a biased estimate for all $t > 0$. Estimating the expected return at every $t$ would require to perform multiple sequence generations for each $s_t$, an approach taken by VinePPO (Kazemnejad et al., 2024) at the cost of a significantly higher compute overhead. Second, $A_{\text{NAE}}$ subtracts the baseline $\mu_t^i$ from the return $G_t^i$, which is used in policy gradient methods to reduce the variance of the policy updates. But this does not provide a parametrized way of balancing variance and bias like GAE does. But, precisely because GRPO uses biased estimates of $V(s_t), \forall t > 0$, it should aim to use generalized advantage estimates. This is the central motivation behind our proposed algorithm, GRPO-$\lambda$, which incorporates a critic-free reformulation of GAE.

**Theorem 1** *The policy gradient estimate $\hat{g}$ using traces from generalized advantage estimation $A_{GAE}$ can be re-parameterized with a critic-free TD-error $\delta_t$ such that $\hat{g} = \sum_{t=0}^{\infty} A_{GAE}(s_t) \nabla_\theta \log \pi_\theta(a_t|s_t) = \sum_{t=0}^{\infty} \delta_t \sum_{l=0}^{t} (\gamma\lambda)^l \nabla_\theta \log \pi_\theta(a_{t-l}|s_{t-l})$. Proof in Appendix A.2.*

Intuitively, Theorem 1 provides an elegant reparameterization of the GAE formulation as weighted cumulative *action* log-probabilities instead of a sum of TD residuals to enable gradient estimation for the language generation setting. The resulting objective function, $\ell_\pi = \min\left( \pi_{\text{ratio}}^{\text{GAE}}(s_t) \delta_t, \text{clip}(\pi_{\text{ratio}}^{\text{GAE}}(s_t), 1 - \epsilon, 1 + \epsilon) \delta_t \right)$, now incorporates GAE's $\lambda$ weighting mechanism in $\pi_{\text{ratio}}^{\text{GAE}}(s_t)$:

$$\pi_{\text{ratio}}^{\text{GAE}}(s_t) = \exp\left( \sum_{l=0}^{t} (\gamma\lambda)^l \log \pi_\theta(a_{t-l}|s_{t-l}) - \sum_{l=0}^{t} (\gamma\lambda)^l \log \pi_{\theta_{\text{old}}}(a_{t-l}|s_{t-l}) \right). \quad (1)$$

Additionally, since we do not have a critic $V_\psi$, we approximate $\delta_t$ using the group returns as in GRPO, i.e., $\delta_t = A_{\text{NAE}}$. Combined with the GAE weighting, this results in GRPO-$\lambda$, which significantly improves the reasoning performance of the resulting post-trained LLM compared to GRPO. GRPO-$\lambda$ is also a generalization of GRPO, as it falls back to GRPO with $\lambda = 0$. Finally, although the computational overhead increases linearly with the sequence length, it is negligible compared to the overall LLM post-training process. In our experiments, we did not notice any significant walltime difference between GRPO and GRPO-$\lambda$.

### 3.1 Bounding the normalized advantage estimation bias

To better understand the bias GRPO introduces by using $V(s_0)$ estimates for states $s_t, \forall\, t > 0$, we analyze the difference between the value in $s_0$ and in $s_t$:

**Lemma 1** *Considering an LLM post-training setting, i.e., a deterministic transition function where $s_t$ is defined by $a_{0:t}$, and a binary reward signal, $\Delta V(s_t) \equiv V(s_0) - V(s_t) \leq 1 - \prod_{k=0}^{t-1} \sum_{a_k \neq a^{EOS}} \pi_\theta(a_k|s_k)$, where $a^{EOS}$ corresponds to the action generating the end-of-sequence token, and thus terminating the episode. Proof in Appendix A.1.*

Intuitively, the probability of generating an EOS token $a^{\mathrm{EOS}}$ increases with time, thus increasing the probability of receiving a positive reward. And so, for a large enough $t$, $\pi_\theta$ might have generated many sequences shorter than $t$. It is those sequences that introduce a bias in GRPO's value estimates. Thus, earlier states have a more accurate estimation of their value.

### 3.2 Alternative weighting mechanisms

The insights provided by Lemma 1 lead us to think more generally about per-token weighting of the policy gradient. Assuming $\gamma = 1$, in our setting, returns and values are the same for each timestep $t$. On one hand, the return for later states have less variance, which allows us to be confident about their gradient updates. On the other hand, early states had more accurate critic estimates, since they are closer to $s_0$. Using our GAE reparameterization as a starting point, we propose reweighting alternatives, that put a different emphasis on a token depending on its position in time.

**Traces as per-token weighting** The discount induced by $\lambda < 1$ results in an exponential decay of weighting importance as we go back in time. Instead of applying it on action log-probabilities, we propose to directly weight $\ell_{\mathrm{NAE}}(s_t)$ with the $\sum_{l=0}^{t}(\gamma\lambda)^l$ trace. This simplifies the problem, as the trace only needs to be computed once, instead of having to sum all the log-probabilities at each policy-update. We refer to this variant as GRPO-$\lambda$ ($\epsilon$-weight). A side-by-side comparison can be found in Appendix A (algorithm 1 and algorithm 2).

**Varying the type of decay** In the RL literature, multiple variations of eligibility traces have been investigated (Williams, 1992; Singh & Sutton, 1996; Seijen & Sutton, 2014; Sutton et al., 2016; van Hasselt et al., 2021) that dictate how they accumulate over time, and thus how much weight they provide at each timestep. Similarly, since our analysis from Lemma 1 indicate two sources of inaccuracies, one on the early tokens, one on the late tokens, we propose a variation of the weighting scheme such that early tokens are considered as important as the late ones:

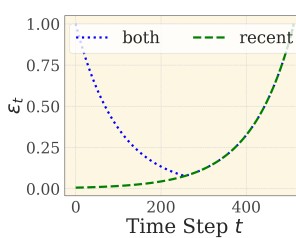

Figure 2: $\epsilon$-trace styles.

$$\pi_{\mathrm{ratio}}^{\mathrm{GAE}}(s_t) = \exp\left(\sum_{l=0}^{t} \mathrm{tr}(t,l) \log \pi_\theta(a_{t-l}|s_{t-l}) - \sum_{l=0}^{t} \mathrm{tr}(t,l) \log \pi_{\theta_{\mathrm{old}}}(a_{t-l}|s_{t-l})\right), \qquad (2)$$

where $\mathrm{tr}(t,l) = \max((\gamma\lambda)^l, (\gamma\lambda)^{t-l})$. The distinction between the classic traces, which we call *recent*, and the proposed variation, which we call *both*, is depicted in Figure 2. We perform extensive experiments and comparisons on all combinations of the different variations, and show that all provide significant improvement over GRPO (see Appendix D), proving that per-token weighting can greatly boost performance for RL finetuning.

## 4 Experiments

**Training details** We do an extensive comparison of our proposed GRPO-$\lambda$ against GRPO with LLMs of diverse sizes (1.5B, 3B and 7B) on mathematic reasoning, similar to related works (Kazemnejad et al., 2024; Roux et al., 2025; Yu et al., 2025; Zhang & Zuo, 2025). We focus on multiple aspects. First, analyze the training efficiency by measuring the increase in average reward on the

training dataset, while maintaining a low KL-divergence between $\pi_{\theta_t}$ and $\pi_{\text{ref}}$. Next, we measure the performance of the final checkpoints of our trained models on multiple challenging mathematic reasoning benchmarks. Finally, to better understand the properties of GRPO-$\lambda$ and the impact of Lemma 1, we perform evaluations on the alternative token weighting mechanisms: *recent* and *both* traces, and *trace* or *weight* token updates. We also assess the choice of $\lambda$, by performing the experiments on our 1.5B models with both $\lambda = 0.99$ and $\lambda = 0.98$. We refer to Appendix B for a full list of hyperparameters, and for the comprehensive information about computational resources to Appendix E.

Specifically, we use `Qwen/Qwen2.5-Math-1.5B-Instruct`, `Deepseekai/Deepseek-R1-Distill-Qwen-1.5B`, `suayptalha/Deepseek-R1-Distill-LLaMA-3B`, and `Deepseekai/Deepseek-R1-Distill-Qwen-7B`. Our RL finetuning pipeline includes an SFT step to train LLMs to reason within a specific format. For the RL finetuning datasets, we use GSM8K Cobbe et al. (2021), Math-12K [1]Lightman et al. (2023), MathRL-16K [2], and ORZ_MATH-57K Hu et al. (2025) which include a variety of challenging math problems. To benchmark, we follow Liu et al. (2025) and evaluate on AIME24 Li et al. (2024), AMC Li et al. (2024), OlympiadBench He et al. (2024), Math500 Hendrycks et al. (2021), and MinervaMath Lewkowycz et al. (2022) benchmarks to report the individual and aggregated performance of the different post-trained LLM checkpoints. For all but the 7B mode, we train across the RL finetuning datasets for 10000 steps. Due to computational limitations, we limit the training of the 7B model to 3500 steps.

## 4.1 ANALYZING THE DIFFERENT TOKEN WEIGHTING SCHEMES

In Section 3.2, we propose alternative token weighting schemes to our re-parameterized general advantage estimation, namely *both, recent* trace weighting styles and $\epsilon$-weight, $\epsilon$-trace token weighting styles. We analyze their effect on the two 1.5B parameter models. Moreover, to better understand the impact of the traces themselves, we incorporate 2 different values for $\lambda$ ($\lambda \in \{0.98, 0.99\}$) in these experiments. Specifically, for each model training on RL finetuning dataset across the different hyper-parameters, we average the performance over the last 100 training steps to understand the effect of different hyper-parameter choices.

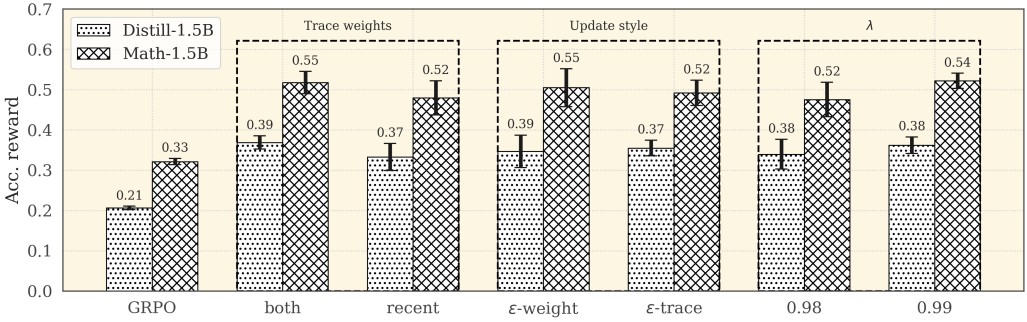

Figure 3: Comparison of the final accuracy reward (smoothened over the last 20 training iterations) for the token weighting schemes, for both 1.5B models trained on the ORZMath57K dataset. Overall, all token weighting schemes improve training accuracy compared the the GRPO baseline. Interestingly, the *both*-style trace weight results in higher performance compared to the classic *recent*-style, showing that alternative token weighting schemes could greatly improve model performance.

First, we observe that the least sensitive hyperparameter is the token weighting style, as both $\epsilon$-weight and $\epsilon$-trace have similar average performance across all datasets. This leads to promising avenues for future work, by providing simple weighting mechanisms that focus on early and late tokens. Despite the similar performance, we stick with $\epsilon$-trace, which is supported theoretically by GAE and aligns with the PPO-style clipped surrogate objective function.

---

[1]https://huggingface.co/datasets/hiyouga/math12k
[2]https://huggingface.co/datasets/riddickz/math-rl-16k

Next, the choice of $\lambda$ affects the back up, and the eligibility of the past states. As $\lambda$ approaches 1, it increases the eligibility for distant state, potentially accelerating the updates. The opposite is true when $\lambda$ approaches 0. We found the performance to be the best at $\lambda = 0.99$ across the different datasets and the two architectures. Finally, for the trace weighting style, *both* systematically outperforms GRPO, and sometimes the classic *recent* style as well. An example of this can be seen in Figure 3, with the other datasets available in Appendix D. Recent work (Bachmann & Nagarajan, 2024) discusses phenomenon where as the sequence progresses, the next-tokens start falling in place thereby making the next token prediction slightly easier. So, while the better performance of weighting style *both* compared to *recent* is interesting from an RL standpoint, the LLM text generation presents not only a convincing explanation but also warrant further investigations for more domain specific and informed credit assignments in RL for LLM scenarios.

Based on the analysis, we pick the best configuration from the experiments across the two models to be ($\lambda$=0.99, weight style=both/recent, update style=$\epsilon$-trace), and ($\lambda$=0.99, weight style=recent, update style=$\epsilon$-trace) for the experiments on the 3B and 7B models respectively.

## 4.2 TRAINING EFFICIENCY

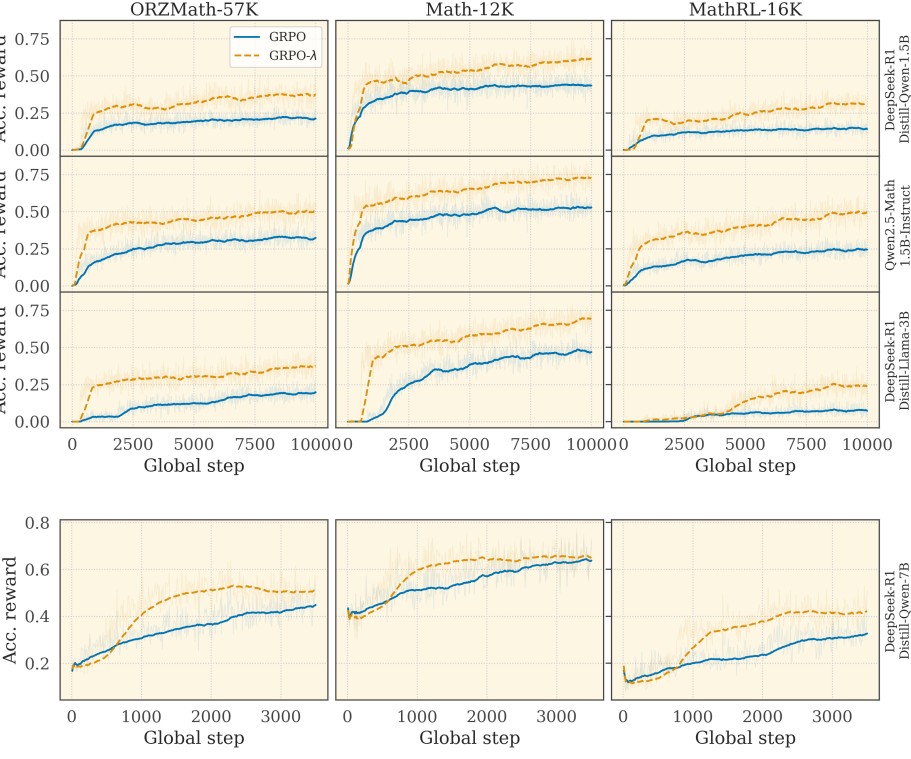

Figure 4: Comparison of training on different RL Math datasets between the best hyperparameter configuration of GRPO-$\lambda$ and GRPO across the 4 different models used throughout this paper (ordered by row). GRPO-$\lambda$ systematically outperforms GRPO in terms of accuracy reward during training.

In Figure 4, we compare the training of GRPO with GRPO-$\lambda$ across the different RL-mathematical reasoning datasets. We excluded the training comparison on GSM8K, for some models include public datasets including GSM8K in their SFT or pretraining corpus thereby affecting the performance [3]. The training plots show a trend of improved training exhibited from using GRPO-$\lambda$ with the average gap between the different variants of GRPO-$\lambda$ and GRPO to be around 20-50%, while the performance itself is affected by the choice of the architectures, and the dataset to post-train.

---

[3] For a complete training comparison across all RL finetuning datasets ref. Appendix C.

For example, the instruct-tuned Qwen2.5-Math-1.5B architecture performs significantly better than Deepseek-R1-Distill-Qwen-1.5B, which is an R1-distilled Qwen2.5-Base model. Likewise, with the size of the architecture the average performance across the methods increases. The 7B model significantly performs better than the 1.5B and 3B models, although, the gap between 3B model and 1.5B models are not very significant. We take this to be an artifact of the model for the base models in Qwen2.5 series have a significantly better performance over LLaMA-3.1 base (Yang et al., 2024; Hui et al., 2024). Depending on the dataset, the difficulty of the sampled mathematical problems varies significantly. For example, post-training on GSM8K results in a higher training performance compared to MathRL-16k or Math12k datasets, as GSM8k's questions are much easier to solve. Also, unsurprisingly the size of the architecture does affect the magnitude of the gains during training for the improvement on 7B model is much lower than on the smaller models. Despite these differences in sizes, datasets and architectures, GRPO-$\lambda$ demonstrates a significant improvement over GRPO through applying the traces for improved reasoning with accelerated update resulting in (a) faster convergence across the models, and (2) improved performance on RL training across smaller architectures.

In addition to the accuracy reward evolution over the steps, we monitor the KL-divergence between the updated policy $\pi_{\theta_t}$ at timestep $t$ and the reference model $\pi_{\text{ref}}$. The accumulated log-probabilities in GRPO-$\lambda$ 's objective function mean their gradients are larger than for GRPO, which increase the risk of deviating from $\pi_{\text{ref}}$. We observe that the KL-divergence stays low throughout training (ref Appendix C), which shows that GRPO-$\lambda$ 's increase in performance is not coming at the cost of overfitting. This is because we adopt two specific techniques to ensure a smooth and stable training:

**Clamping the advantage function** For fine-tuning LLMs with RL, Roux et al. (2025) have observed that positive and negative returns in the policy gradient loss produce drastically different behavior in terms of gradient updates. Negative returns encourage $\pi$ to move as far as possible from the corresponding trajectory, which can act as a destructive force on model parameters. GRPO-$\lambda$ multiplies advantages instead of returns with $\log \pi_\theta(a_t, s_t)$, but, as-is, we observe similar trends as Roux et al.'s observations. To mitigate this issue, we adopt a similar approach, i.e., we clamp negative advantages to a small value ($-0.1$ in our experiments). With the proposed clamping, the KL-divergence is stable, albeit higher than GRPO. We argue that a higher - but stable - KL-divergence may in fact improve learning, as a too strong KL regularization potentially limits exploration during policy optimization (Hu et al., 2025; Zhang & Zuo, 2025), and a high regularization term ($\beta$) does not correlate with better learning (Lambert et al., 2024).

**Mitigate reward hacking:** We observed during training of GRPO-$\lambda$ on non-SFT'ed LLMs (Qwen2.5-Math-1.5B-Instruct) with the objective of optimizing both the "format" and "accuracy" of the response generated may lead to an unstable training, where the LLM learns to hack the reward functions to end up optimizing for the easier reward functions after a long number of steps Skalse et al. (2022). To avoid this behavior, we train LLM with single reward RL, to maximize the accuracy reward, and a pre-SFT step to improve formatting. We observe the format reward to stay high throughout the RL training without forgetting the formatting learnt in the SFT step.

The results across two different architectures (LLaMA3.1, and Qwen2.5) and different sizes 1.5B, 3B and 7B on 4 different training datasets demonstrate that GRPO-$\lambda$ is indeed stable and is much better than GRPO to train better on RL datasets through credit assignment.

### 4.3 BENCHMARK PERFORMANCE

In Table 1, we compare the performance of LLMs post-trained with GRPO and GRPO-$\lambda$ on different train-datasets across 5 challenging and popular math-reasoning benchmarks, AIME24, AMC, OlympiaBench, Math500 and MinervaMath. We observe that the average improvement that GRPO-$\lambda$ has over GRPO is quite significant. However, the choice of the dataset to post-train appears to have an effect on the benchmark performance.

First, models that have been trained on the ORZMath57K perform far worse on the evaluation tasks than the models trained on other benchmarks. This is consistent across multiple model architectures and sizes, be it for GRPO and for GRPO-$\lambda$ . Upon further investigation, we found that these models are much less accurate in providing an answer in the valid format. The effect of different datasets on the RL finetuning warrants a special treatment, which, however, is out of scope for this paper. Next,

Table 1: Average performance for each evaluation benchmark across the different training datasets. This table only contains GRPO-$\lambda$ , not the variations introduced in subsection 3.2.

| Model | Method | Evaluation benchmark (accuracy) | | | | | |
|---|---|---|---|---|---|---|---|
| | | Average | AIME24 | AMC | MATH | Minerva | Olympiad |
| Qwen-1.5B | GRPO | 0.334 | 0.067 | 0.380 | 0.686 | 0.219 | 0.318 |
| | GRPO-$\lambda$ | 0.346 | 0.104 | 0.381 | 0.699 | 0.215 | 0.333 |
| R1-Distill-Qwen-1.5B | GRPO | 0.335 | 0.117 | 0.416 | 0.675 | 0.191 | 0.278 |
| | GRPO-$\lambda$ | 0.363 | 0.142 | 0.443 | 0.716 | 0.212 | 0.303 |
| R1-Distill-LLaMA-3B | GRPO | 0.142 | 0.042 | 0.092 | 0.309 | 0.114 | 0.092 |
| | GRPO-$\lambda$ | 0.200 | 0.034 | 0.202 | 0.450 | 0.172 | 0.144 |
| R1-Distill-Qwen-7B | GRPO | 0.429 | 0.200 | 0.491 | 0.775 | 0.320 | 0.361 |
| | GRPO-$\lambda$ | 0.451 | 0.217 | 0.518 | 0.800 | 0.334 | 0.384 |

we observe that the 3B model performs worse on the evaluation benchmarks than the smaller 1.5B models. We believe this is due to the fact that the 3B model is the only one using a Llama architecture, while the other ones use Qwen2.5, which generally performs better than Llama3.1 on mathematical tasks (Yang et al., 2024). Finally, post-training on the MathRL-16k dataset results in particularly good performance on the evaluation benchmarks for both GRPO and GRPO-$\lambda$ , with GRPO-$\lambda$ resulting in an improvement of over 5 points (0.0552) across the benchmarks. This impressively leads to the 7B model post-trained with GRPO-$\lambda$ producing a correct answer $47.6\%$ of the time, as shown in Figure 1. We refer to Appendix E for complete benchmarking performance of all the checkpoints.

## 5 LIMITATIONS

While fine-tuning a LLM with GRPO-$\lambda$ greatly improve its training and evaluation performance compared to GRPO, it comes at the expense of a decrease in training stability. The KL-divergence between GRPO-$\lambda$ 's fine-tuned $\pi_\theta$ and $\pi_{\text{ref}}$ is larger than for GRPO, and becomes order of magnitudes larger with the negative advantage clamping. Moreover, even though we derive a bound on the bias of using $V(s_0)$ instead of $V(s_t)$, explicitly incorporating this bias into the advantage estimation was detrimental to the policy improvement (see related experiments in Appendix E, and more extended discussion in subsection A.2). This seems to indicate that GRPO-$\lambda$ 's objective function is quite sensitive. Additionally, all the experiments we performed are focused on mathematical reasoning datasets. It is unknown if we will also witness the same gain in performance we have seen on these benchmarks on other reasoning tasks, such as coding, or even general-purpose tasks. Finally, even though Table 1 indicates that the improvement gap between GRPO-$\lambda$ and GRPO is larger on the 3B and 7B models than the 1.5B models, it remains to be seen if this improvement gap scales to models with even more parameters, e.g., the 32B or 72B variants of Qwen2.5, due to computation restrictions.

## 6 CONCLUSION AND FUTURE WORK

We present GRPO-$\lambda$ , which significantly outperforms GRPO both in terms of training and evaluation performance across 4 training datasets, 5 mathematical reasoning benchmarks, 3 different model sizes and 2 different model architectures. In contrast to GRPO, GRPO-$\lambda$ incorporates a reformulation of the generalized advantage estimation, allowing it to rapidly back-propagate the sequence reward to relevant tokens. We show that GRPO's advantage term increasingly biases value estimates for later tokens, spurring us to investigate alternative token weighting schemes that put a higher focus on early tokens. Extensive experiments show that this can result in higher performance than the classic traces used by GAE. This leads to an interesting avenue for future work, i.e., by analyzing the impact of different types of traces. Moreover, even though our experiments when incorporating the bound directly into the advantage estimate were inconclusive, we believe it warrants further investigation, to not only improve the accuracy of the updates, but also better stabilize GRPO-$\lambda$ 's training.

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

# A  GRPO-$\lambda$ Theory

We propose a re-parametrization of the generalized advantage estimation so they can be taken advantage of by critic-free methods such as GRPO.

## A.1  Full Version of Lemma 1

First, we take a look at GRPO's advantage estimation, which is centered around producing multiple rollouts from the start-state $s_0$, and using their resulting scores to estimate $\mu_t^i$, the mean return of all states $\left\{ s_t^0, \ldots, s_t^{g-1} \right\}$ in group $\mathcal{G}$.

The mean $\mu_t^i$ serves as a baseline to reduce the variance of $G_t$ in PPO's policy update. Indeed, a learned function of the state $b(s_t)$ (where typically $b(s_t) = V_\pi(s_t)$) can be used as a baseline, keeping the policy gradient unbiased. However, $\mu_t^i$ does not really depend on $s_t$, as it is an average of returns starting from $s_0$. This difference $\Delta V(s_t) \equiv V(s_0) - V(s_t)$ and the bias it introduces is what we analyze in the following Lemma.

**Lemma 1** *Considering an LLM post-training setting, i.e., a deterministic transition function where $s_t$ is defined by $a_{0:t}$, and a binary reward signal, $\Delta V(s_t) \equiv V(s_0) - V(s_t) \leq 1 - \prod_{k=0}^{t-1} \sum_{a_k \neq a^{EOS}} \pi_\theta(a_k|s_k)$, where $a^{EOS}$ corresponds to the action generating the end-of-sequence token, and thus terminating the episode.*

**Proof:**  By definition, the value $V(s)$ is the expectation over the returns from any given state $s$. Thus, $V(s_0)$ can be written as,

$$V_\pi(s_0) = \mathbb{E}_\pi \left[ G_0 \right] = \mathbb{E}_\pi \left[ \sum_{t=0}^{T} \gamma^t r_t \right]$$

The sum until $T$ can be split into two terms, $0 - t$, and $t - T$. Then,

$$\begin{aligned} V_\pi(s_0) &= \mathbb{E} \left[ \sum_{k=0}^{t-1} \gamma^k r_k + \sum_{k=t}^{T} \gamma^k r_k \right] \\ &= \mathbb{E} \left[ \sum_{k=0}^{t-1} \gamma^k r_k \right] + \gamma^t \mathbb{E} \left[ \sum_{k=t}^{T} \gamma^{k-t} r_k \right]. \end{aligned} \tag{3}$$

The second term is the expected return from $s_t$, which is nothing but $V_\pi(s_t)$. Then,

$$V(s_0) = \mathbb{E} \left[ \sum_{k=0}^{t-1} \gamma^k r_k \right] + \gamma^t V(s_t),$$

$$\begin{aligned} \Delta V(s_t) &= V(s_0) - V(s_t) \\ &\leq V(s_0) - \gamma^t V(s_t) \\ &\leq \mathbb{E} \left[ \sum_{k=0}^{t-1} \gamma^k r_k \right]. \end{aligned} \tag{4}$$

In our LLM post-training setting, all intermediate rewards are zero, i.e., $r_t = 0, \forall t < T$. When an EOS-token is selected as action, the reward $r_T$ is either 1 (the policy provided the correct answer to the problem) or 0 (the policy provided an incorrect answer). Thus, $V(s_t) \geq 0, \forall s_t$. More interestingly, $\mathbb{E} \left[ \sum_{k=0}^{t-1} \gamma^k r_k \right] > 0$ only if there exist cases where the episode ends before timestep $t$.

Assuming every episode is successful, i.e., every EOS-token yields a reward of 1, then $\mathbb{E}\left[\sum_{k=0}^{t-1} \gamma^k r_k\right]$ is upper-bounded by the probability of generating at least one EOS-token at any time during an episode. Thus,

$$\Delta V(s_t) \leq \mathbb{E}\left[\sum_{k=0}^{t-1} \gamma^k r_k\right] \leq 1 - \underbrace{\prod_{k=0}^{t-1} \sum_{a_k \neq a^{\text{EOS}}} \pi_\theta(a_k|s_k)}_{\substack{\text{probability of not generating} \\ \text{at least one EOS-token in } t \text{ timesteps}}} \quad . \tag{5}$$

■

## A.2 FULL VERSION OF THEOREM 1

Next, we repurpose the bias-variance trade-off from GAE towards a critic-free policy-update like GRPO.

**Theorem 1** *The policy gradient estimate $\hat{g}$ using traces from generalized advantage estimation $A_{GAE}$ can be re-parameterized with a critic-free TD-error $\delta_t$ such that $\hat{g} = \sum_{t=0}^{\infty} A_{GAE}(s_t)\nabla_\theta \log \pi_\theta(a_t|s_t) = \sum_{t=0}^{\infty} \delta_t \sum_{l=0}^{t}(\gamma\lambda)^l \nabla_\theta \log \pi_\theta(a_{t-l}|s_{t-l}).$*

**Proof:** We reorganize the many terms of the policy gradient formula[4] so that the gradient is of the form: $g = \mathbb{E}_\pi\left[\sum_{t=0}^{\infty} r_t \psi_t\right]$, where $\psi_t$ can depend only on the states, actions, and rewards that occurred before (or immediately after) the arrival of $r_t$. We will approximate for an online estimator of the policy gradient $\hat{g}$:

$$\hat{g} = \sum_{t=0}^{\infty} A_{\text{GAE}}(s_t)\, \nabla_\theta \log\, \pi_\theta(a_t \mid s_t) \tag{6}$$

$$= \sum_{t=0}^{\infty} \nabla_\theta \log\, \pi_\theta(a_t \mid s_t)\, \sum_{l=0}^{\infty}(\gamma\lambda)^l \delta_{t+l}^V \tag{7}$$

Let us introduce the following shorthand:

$$\nabla_t := \nabla_\theta \log\, \pi_\theta(a_t \mid s_t), \tag{8}$$

then expand the sum:

$$\begin{aligned}
\hat{g} = & \nabla_0(\delta_0^V + (\gamma\lambda)\delta_1^V + (\gamma\lambda)^2\delta_2^V + \dots) \\
& + \nabla_1(\delta_1^V + (\gamma\lambda)\delta_2^V + (\gamma\lambda)^2\delta_3^V + \dots) \\
& + \nabla_2(\delta_2^V + (\gamma\lambda)\delta_3^V + (\gamma\lambda)^2\delta_4^V + \dots) \\
& + \dots
\end{aligned} \tag{9}$$

group the $\delta_t^V$ terms:

$$\begin{aligned}
\hat{g} = & \delta_0^V \nabla_0 \\
& + \delta_1^V (\nabla_1 + (\gamma\lambda)\nabla_0) \\
& + \delta_2^V (\nabla_2 + (\gamma\lambda)\nabla_1 + (\gamma\lambda)^2\nabla_0) \\
& + \dots
\end{aligned} \tag{10}$$

---

[4]With the existence of REINFORCE Williams (1992) and policy gradient methods, several works (e.g., Sun et al. (2018)) have used the reformulation of the policy gradients under different settings.

and summarize:

$$\hat{g} = \sum_{t=0}^{\infty} \delta_t^V \sum_{l=0}^{t} (\gamma\lambda)^l \nabla_{t-l} \tag{11}$$

$$= \sum_{t=0}^{\infty} \delta_t^V \sum_{l=0}^{t} (\gamma\lambda)^l \nabla_\theta \log \pi_\theta(a_{t-l} \mid s_{t-l}) \tag{12}$$

Moreover, by defining eligibility trace as the inner sum in that equation:

$$\epsilon_t := \sum_{l=0}^{t} (\gamma\lambda)^l \nabla_{t-l} \tag{13}$$

and converting to a recursive formula:

$$\epsilon_0 := \nabla_0 \tag{14}$$
$$\epsilon_t := (\gamma\lambda)\epsilon_{t-1} + \nabla_t, \tag{15}$$

we have our online generalized advantage estimator for the policy gradient:

$$\hat{g} = \sum_{t=0}^{\infty} \delta_t^V \epsilon_t \tag{16}$$

So at each time-step, we compute the gradient term $\hat{g}_t = \delta_t^V \epsilon_t$ as the product of the TD error. The role of $\lambda \in [0, 1]$ is unchanged, remaining a bias-variance trade-off. For $\lambda = 0$, the problem reduces to the (unbiased) TD(0) function. As we increase $\lambda$ towards 1, we reduce the variance of our estimator but increase the bias.

∎

**Relation with GRPO's normalized advantage estimation (NAE)**   In this work, we consider $\delta_t^V = A_{\text{NAE}}(s_t)$. The advantage is often used in policy-gradient methods Mnih et al. (2016), where subtracting the value-estimate of the current state is used as a variance-reduction technique:

$$\delta_t^V = G_t - V(s_t).$$

However, $A_{\text{NAE}}(s_t)$ subtracts $V(s_0)$ instead of $V(s_t)$. Using $\Delta V(s_t) \equiv V(s_0) - V(s_t)$ introduced in Lemma 1 results in:

$$\delta_t^V = G_t - V(s_0) + \Delta V(s_t).$$

We can thus potentially reduce the bias from using $V(s_0)$ by including knowledge about $\Delta V(s_t)$ in $\delta_t^V$.

In preliminary experiments, we simply added the upper bound for $\Delta V(s_t)$ to $A_{\text{NAE}}(s_t)$.

However, results were inconclusive. The bias bound assumes that the policy is optimal, resulting in the highest potential value for $\Delta V$. GRPO removes this term, resulting in a pessimistic advantage value (compared to having an actual critic). By naively adding the upper bound of $\Delta V$, we surmise this might be over-correcting the bias term, resulting in an optimistic advantage value. This means that including the bound could reinforce action-probabilities that lead to 0-rewards (encouraging false positives) while, in the case of GRPO, it instead reduces action-probabilities that lead to 1-rewards (penalizing the false negatives). In the former case, we are stuck in suboptimal behavior, while in

the latter, there could be alternative paths towards solving the problem. As such, a more careful bias correction term is needed. It is important to mention that this only focuses on $\Delta_V$ in the GRPO setup, not GRPO-$\lambda$ . Adding eligibility traces balances the bias-variance trade-off, which is one of the reasons why GRPO-$\lambda$ outperforms GRPO. We aim to further this direction in future work.

### A.3 PSEUDOCODE

In section 3, we introduce GRPO-$\lambda$ , the variant $\epsilon$-trace follows the traditional RL literature and applies the trace before the PPO clipping objective. algorithm 1 (left) represents GRPO-$\lambda$ , with the traces derived from Theorem 1. Likewise, the trace-inspired $\epsilon$-weight variant is illustrated in algorithm 2. We color code the pseudocodes where blue denotes the modifications on the GRPO algorithm done for the $\epsilon$-trace variant of the GRPO-$\lambda$ and red denote the modifications for the $\epsilon$-weight variant.

A less rigorous explanation of $\epsilon$-weight variant is that the loss at each step (token) essentially gets reweighted implicitly with $\epsilon$-trace. $\epsilon$-weight does this reweighting of the loss explicitly. To that, this uses the PPO-clip to provide the td-error, $\delta^V$. Multiplying the trace weights with $\log \pi_{x_r^i}^\theta$ provides the traces, $\epsilon_{x_r^i}^w$, which then multiplied with the advantage estimated explicitly weighs the loss of different tokens ($\epsilon_{x_r^i}^w A_{x_r^i}^\theta \log \pi_\theta$). However, we observe multiplying that $\log \pi_{x_r^i}^\theta$ results in instability. We alleviate that by soft clamping $\log \pi_{x_r^r}^\theta$ with $1 + \sigma(\log \pi_{x_i^i}^\theta - 1)$. The choice of the clamping function, $f$, sigmoid can be replaced with tanh. The clamping function, $f$, (a) ensures linear dependence on the gradient and preserve the spirit of the traces, and (b) acts as a regularization to prevent extreme values.

---

**Algorithm 1:** GRPO-$\lambda$ ($\epsilon$-trace)

**Input:** $G, \pi^\theta, \pi^{ref}, \gamma, \epsilon_{\text{style}}, \epsilon_{\text{clip}}, \beta, A_{\text{NAE}}$
**for** $i \in |G|$ **do**
  $\hat{A}_{\text{NAE}} \leftarrow \max(A_{\text{NAE}}, -0.1)$;
  $\text{coef}_1 \leftarrow$

$$\exp(\sum_{l=0}^{t}(\gamma\lambda)^l \log \pi_{x_{t-l}^i}^\theta -$$

$$\sum_{l=0}^{t}(\gamma\lambda)^l \log \pi_{x_{t-l}^i}^{old})$$

  $\text{coef}_2 \leftarrow \text{clamp}(\text{coef}_1, 1 \pm \epsilon_{\text{clip}})$;
  $\delta_{x_r^i}^V \leftarrow \hat{A}_{\text{NAE}}(x_r^i)$;
  $\ell_{x_t^i}^{\text{GRPO-}\lambda} \leftarrow -\min(\text{coef}_1, \text{coef}_2)\delta_{x_r^i}^V$;
  $\ell_{\text{GRPO-}\lambda} \leftarrow \ell_{\text{GRPO-}\lambda} + \beta \cdot \ell_{KL}$;

---

**Algorithm 2:** GRPO-$\lambda$ ($\epsilon$-weight)

**Input:** $G, \pi^\theta, \pi^{ref}, \gamma, \epsilon_{\text{style}}, \epsilon_{\text{clip}}, \beta, A_{\text{NAE}}$
**for** $i \in |G|$ **do**
  $\hat{A}_{\text{NAE}} \leftarrow \max(A_{\text{NAE}}, -0.1)$;
  $\text{coef}_1 \leftarrow \exp(\log \pi_{x_t^i}^\theta - \log \pi_{x_t^i}^{old})$;
  $\text{coef}_2 \leftarrow \text{clamp}(\text{coef}_1, 1 \pm \epsilon_{clip})$;
  $w_t \leftarrow \sum_{l=0}^{t}(\gamma\lambda)^l$;
  $A_{x_r^i}^{\pi_\theta} \leftarrow -\min(\text{coef}_1, \text{coef}_2)\hat{A}_{\text{NAE}}(x_r^i)$;
  $\epsilon_{x_r^i}^w \leftarrow w @ (1 + \sigma(\log \pi_{x_r^i}^\theta - 1))$;
  $\ell_{\text{GRPO-}\lambda} \leftarrow \epsilon_{x_r^i}^w A_{x_r^i}^{\pi_\theta}$;
  $\ell_{\text{GRPO-}\lambda} \leftarrow \ell_{\text{GRPO-}\lambda} + \beta \cdot \ell_{KL}$

---

**Upper bound $\Delta V_t$** The estimation of $\delta^V$ by Theorem 1 requires an upper bound for $\Delta V_t$. In Lemma 1 we derive the upper bound to be the probability of generating an EOS-token. Our implementation does not use this upper bound in its advantage.

**Trace weights** Trace matrix is a non-learnable precomputed lower-triangular matrix with 1s on the leading diagonal. The two variants of GRPO-$\lambda$ uses *recent* and *both* styles for the trace-weights, which is computed with get_trace(). The get_trace() method takes in as arguments: ($\gamma$, $\lambda$, max_length, style: both, recent). For the choice *both*, the trace matrix is estimated as:

$$\text{trace}^{both} = \begin{cases} 1, & \text{if rows} = \text{cols} \\ \max\left(\max\left(\epsilon, (\gamma\lambda)^{n-\text{cols}}\right), (\gamma\lambda)^{\text{cols}}\right), & \text{otherwise} \end{cases}, \qquad (17)$$

for *recent*:

$$\text{trace}^{recent} = \begin{cases} 1, & \text{if rows} = \text{cols} \\ \max\left(\epsilon,\ (\gamma\lambda)^{\max(\text{cols})-\text{cols}}\right), & \text{otherwise} \end{cases} \tag{18}$$

Usage of weight style *both* as an alternate to *recent*, and the strong training and benchmark performance that this provides is encouraging and serves as a precursor to explore alternate weighting styles that are domain or data dependent.

## B  Hyperparameters

This section provides a complete overview of all hyperparameters used to run our experiments. Our codebase is based on Huggingface OpenR1.

For evaluation, we use the understanding-r1-zero codebase.

Table 2: Hyperparameter configurations used for GRPO-$\lambda$ .

| Model size | 1.5B | 3B | 7B |
|---|---|---|---|
| Precision | | bf16 | |
| Distributed type | | Deepspeed | |
| Number of devices | | 4 | |
| **Supervised Finetuning stage** | | | |
| Epochs | | 1 | |
| Max sequence length | | 4096 | |
| Learning rate | | $2 \times 10^{-5}$ | |
| Per device batch-size | | 2 | |
| Gradient accumulation steps | | 8 | |
| Full finetuning (no LoRA) | yes | yes | no |
| **GRPO and GRPO-$\lambda$ configuration** | | | |
| Training steps | | 10000 | 3500 |
| Maximum prompt length | | 256 | |
| Group size (number of generations) | | 8 | |
| Maximum gradient norm | | 1.0 | |
| KL-divergence coefficient ($\beta$) | | 0.04 | |
| Accuracy reward weight | | 1.0 | |
| Format reward weight | | 0.0 | |
| Maximum completion length | 256 | 256 | 768 |
| Per device batch-size | 16 | 16 | 8 |
| Gradient accumulation steps | 1 | 1 | 2 |
| Full finetuning (no LoRA) | yes | yes | no |
| **GRPO-$\lambda$ specific configuration** | | | |
| Advantage clamping | | $-0.1$ | |

### B.1  Computational resources

The performed experiments were executed on a High Performance Computing (HPC) cluster comprised of 42 nodes, each containing 4 NVIDIA H100 GPUs, 2 Intel Xeon Gold 6442Y CPUs, and 512GB memory. Each experiment required one node (4 H100 GPUs), with 1.5B and 3B models running for 16 hours (full-finetuning, with no gradient accumulation), and the 7B model running for 40 hours (LoRA finetuning, with gradient accumulation of 2). For each 1.5B, 3B, 7B parameter model, we performed 1 baseline experiment (GRPO) and 8, 2, 1 variations of GRPO-$\lambda$ , respectively, on 4 different datasets. This resulted in a total of 250 GPU-days.

## C  TRAINING PLOTS

This section provides the training plots of all the experiments performed in this work. We show that, regardless of the value for $\lambda$, the weighting update ($\epsilon$-trace, $\epsilon$-weight) or the type of trace (both, recent) used, GRPO-$\lambda$ has a higher training accuracy than GRPO.

Additionally, we show the plots comparing the KL-divergence between $\pi_\theta$ and $\pi_{\text{ref}}$. While the KL-divergence is higher for GRPO-$\lambda$ than for GRPO, it remains quite stable over the training duration. The crucial aspect to avoid explosion of KL-divergence is the clamping of negative advantages to $-0.1$, inspired by Roux et al. (2025).

## C.1 QWEN2.5-MATH-1.5B-INSTRUCT

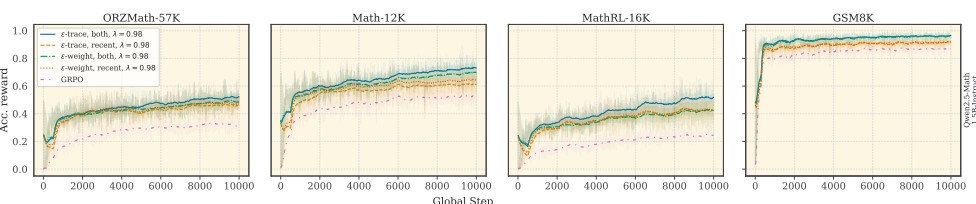

(a) Comparison of training reward over time across all hyperparameters with $\lambda = 0.98$ on Qwen2.5-Math-Instruct-1.5B.

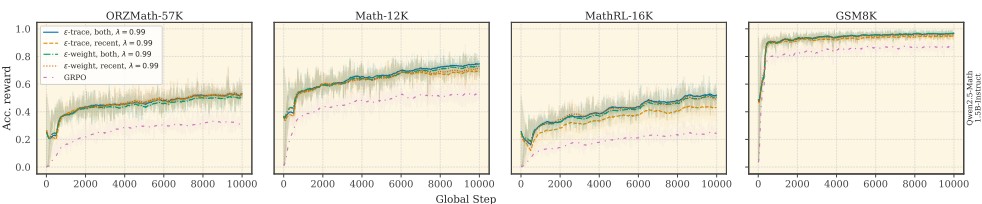

(b) Comparison of training reward over time across all hyperparameters with $\lambda = 0.99$ on Qwen2.5-Math-Instruct-1.5B.

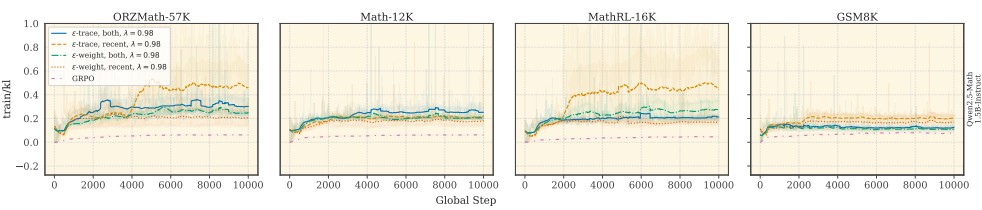

(a) Comparison of KL$(\theta_t \| \theta_{ref})$ across all hyperparameters with $\lambda = 0.98$ on Qwen2.5-Math-Instruct-1.5B.

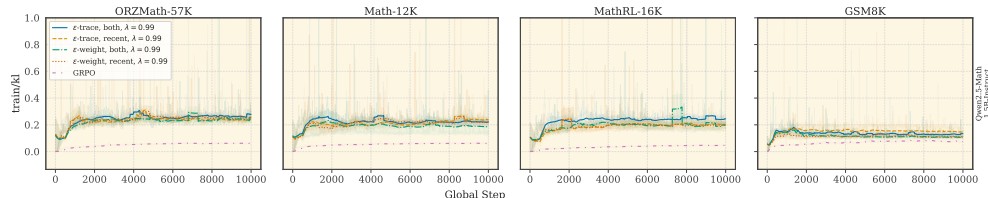

(b) Comparison of KL$(\theta_t \| \theta_{ref})$ across all hyperparameters with $\lambda = 0.99$ on Qwen2.5-Math-Instruct-1.5B.

## C.2 DEEPSEEK-R1-DISTILL-QWEN-1.5B

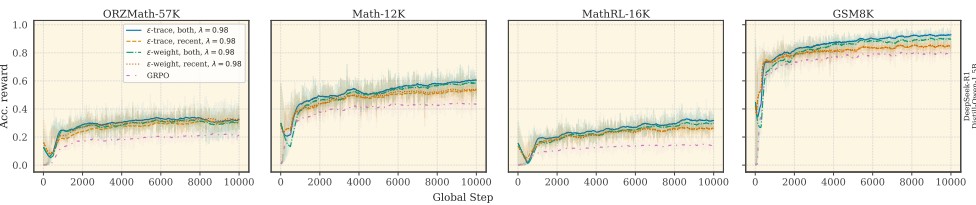

(a) Comparison of training reward over time across all hyperparameters with $\lambda = 0.98$ on DeepSeek-R1-Distill-Qwen-1.5B.

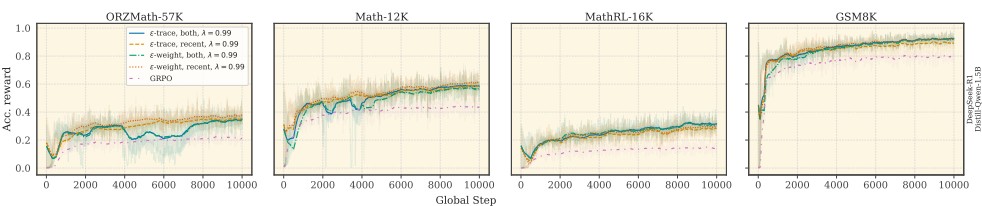

(b) Comparison of training reward over time across all hyperparameters with $\lambda = 0.99$ on DeepSeek-R1-Distill-Qwen-1.5B.

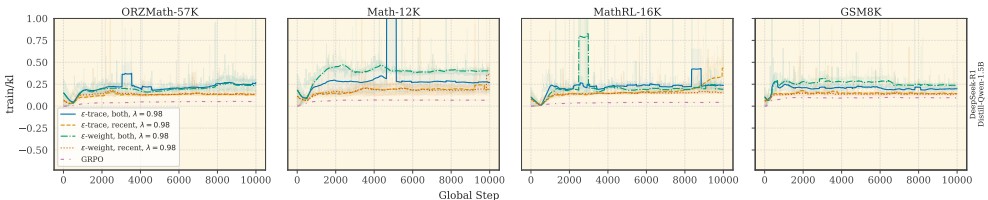

(a) Comparison of $\text{KL}(\theta_t || \theta_{ref})$ across all hyperparameters with $\lambda = 0.98$ on DeepSeek-R1-Distill-Qwen-1.5B.

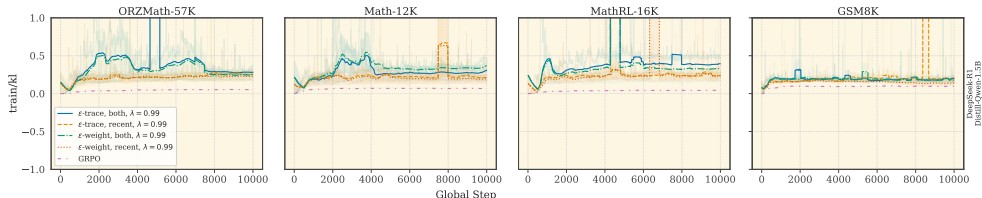

(b) Comparison of $\text{KL}(\theta_t || \theta_{ref})$ across all hyperparameters with $\lambda = 0.99$ on DeepSeek-R1-Distill-Qwen-1.5B.

## C.3 DEEPSEEK-R1-DISTILL-LLaMA-3B

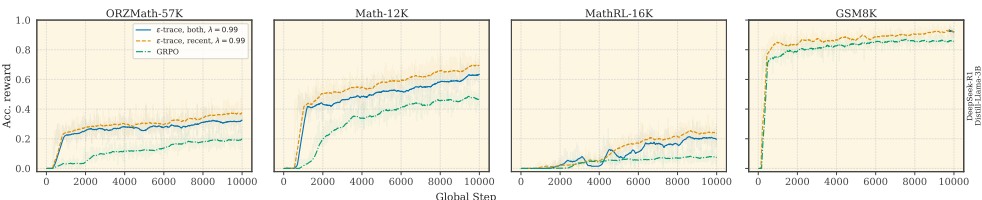

(a) Comparison of training reward over time across all hyperparameters with $\lambda = 0.99$ on R1-Distill-Llama-3B.

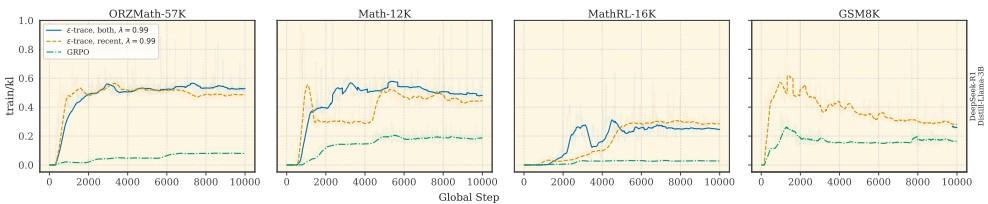

(b) Comparison of $\text{KL}(\theta_t || \theta_{ref})$ across all hyperparameters with $\lambda = 0.99$ on DeepSeek-R1-Distill-Llama-3B.

## C.4 DEEPSEEK-R1-DISTILL-QWEN-7B

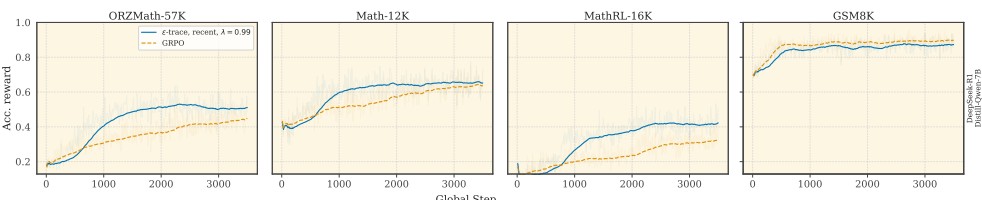

(a) Comparison of training reward over time across all hyperparameters with $\lambda = 0.99$ on DeepSeek-R1-Distill-Qwen-7B.

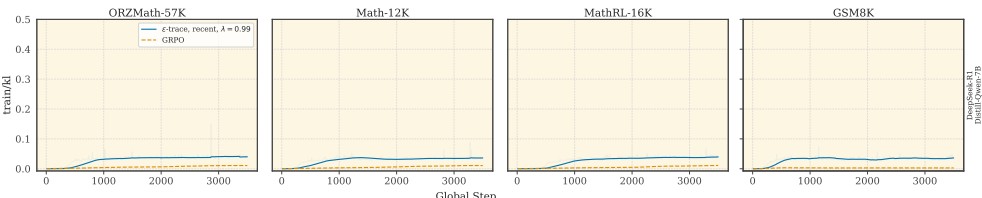

(b) Comparison of $\text{KL}(\theta_t||\theta_{ref})$ across all hyperparameters with $\lambda = 0.99$ on DeepSeek-R1-Distill-Qwen-7B.

# D HYPER-PARAMETER COMPARISONS

Figure 3 in the main manuscript shows the final training performance for each variation of each 1.5B parameter model, using the ORZMath57K dataset for RL post-training. Here, we show the same training performance plots for the models post-trained on the other datasets. The observations made in subsection 4.1 remain valid for the other datasets.

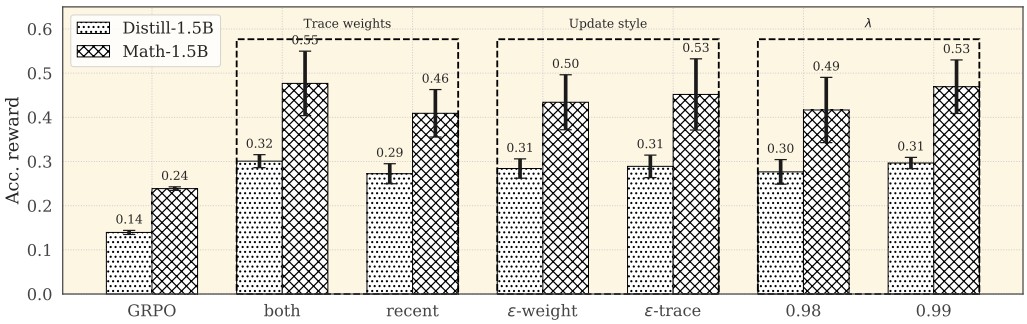

(a) HP comparison for MathRL16K

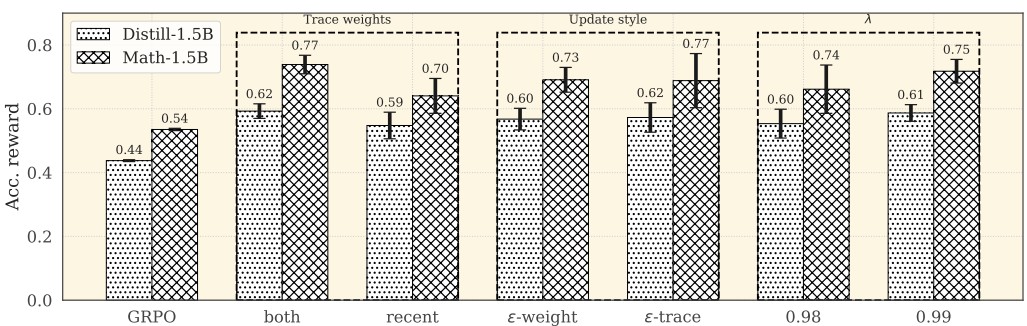

(b) HP comparison for Math12K

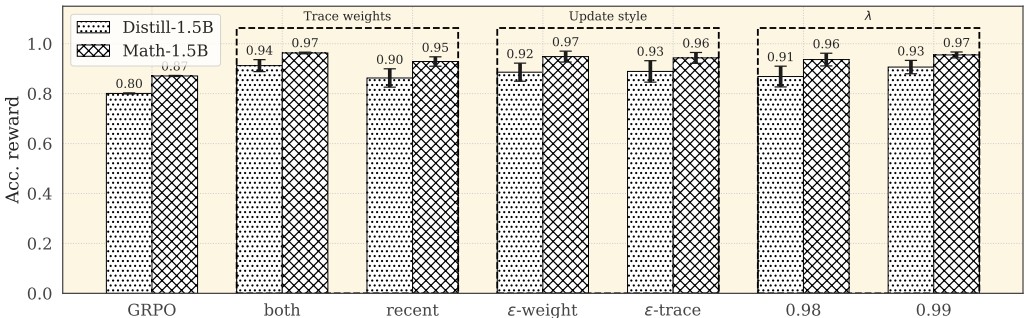

(c) HP comparison for GSM8K

# E COMPLETE BENCHMARK RESULTS

## E.1 EVALUATION SCORES

Table 3: Benchmark results for Qwen2.5-Math-1.5B-Instruct

| Trace Style | GRPO | | | both | | | | recent | |
| Update Style | GRPO | $\epsilon$-trace | | $\epsilon$-weight | | $\epsilon$-trace | | $\epsilon$-weight | |
| $\lambda$ | GRPO | 0.98 | 0.99 | 0.98 | 0.99 | 0.98 | 0.99 | 0.98 | 0.99 |
|---|---|---|---|---|---|---|---|---|---|
| *Trained on math12k* | | | | | | | | | |
| AIME | 0.067 | 0.067 | 0.133 | 0.100 | 0.033 | 0.067 | 0.133 | 0.100 | 0.100 |
| MATH | 0.704 | 0.744 | 0.756 | 0.746 | 0.682 | 0.720 | 0.688 | 0.718 | 0.696 |
| AMC | 0.386 | 0.458 | 0.470 | 0.410 | 0.410 | 0.410 | 0.422 | 0.361 | 0.337 |
| OLBen | 0.324 | 0.348 | 0.366 | 0.360 | 0.317 | 0.327 | 0.317 | 0.336 | 0.301 |
| MIN | 0.191 | 0.206 | 0.217 | 0.199 | 0.221 | 0.213 | 0.195 | 0.232 | 0.213 |
| AVG | *0.334* | 0.365 | **0.388** | 0.363 | 0.333 | 0.347 | 0.351 | 0.349 | 0.329 |
| *Trained on math-rl-16k* | | | | | | | | | |
| AIME | 0.067 | 0.100 | 0.133 | 0.067 | 0.200 | 0.100 | 0.167 | 0.033 | 0.100 |
| MATH | 0.652 | 0.684 | 0.682 | 0.710 | 0.696 | 0.692 | 0.704 | 0.626 | 0.712 |
| AMC | 0.325 | 0.325 | 0.410 | 0.373 | 0.506 | 0.313 | 0.325 | 0.337 | 0.434 |
| OLBen | 0.293 | 0.293 | 0.323 | 0.307 | 0.366 | 0.323 | 0.321 | 0.277 | 0.332 |
| MIN | 0.202 | 0.228 | 0.191 | 0.206 | 0.246 | 0.217 | 0.199 | 0.173 | 0.206 |
| AVG | *0.308* | 0.326 | 0.348 | 0.333 | **0.403** | 0.329 | 0.343 | 0.289 | 0.357 |
| *Trained on gsm8k* | | | | | | | | | |
| AIME | 0.067 | 0.100 | 0.067 | 0.200 | 0.167 | 0.067 | 0.133 | 0.100 | 0.100 |
| MATH | 0.706 | 0.666 | 0.716 | 0.716 | 0.706 | 0.684 | 0.704 | 0.710 | 0.718 |
| AMC | 0.422 | 0.337 | 0.398 | 0.410 | 0.410 | 0.422 | 0.398 | 0.410 | 0.422 |
| OLBen | 0.345 | 0.319 | 0.344 | 0.369 | 0.356 | 0.348 | 0.350 | 0.338 | 0.335 |
| MIN | 0.257 | 0.188 | 0.213 | 0.213 | 0.202 | 0.254 | 0.217 | 0.224 | 0.228 |
| AVG | *0.359* | 0.322 | 0.347 | **0.382** | 0.368 | 0.355 | 0.360 | 0.356 | 0.360 |
| *Trained on orzmath57k* | | | | | | | | | |
| AIME | 0.067 | 0.067 | 0.133 | 0.067 | 0.133 | 0.100 | 0.067 | 0.033 | 0.033 |
| MATH | 0.682 | 0.630 | 0.686 | 0.716 | 0.736 | 0.692 | 0.708 | 0.606 | 0.684 |
| AMC | 0.386 | 0.386 | 0.350 | 0.398 | 0.446 | 0.386 | 0.373 | 0.386 | 0.386 |
| OLBen | 0.311 | 0.244 | 0.292 | 0.333 | 0.333 | 0.335 | 0.341 | 0.262 | 0.324 |
| MIN | 0.226 | 0.254 | 0.202 | 0.202 | 0.210 | 0.228 | 0.195 | 0.151 | 0.217 |
| AVG | *0.334* | 0.316 | 0.332 | 0.343 | **0.378** | 0.348 | 0.337 | 0.276 | 0.323 |

Table 4: Benchmark results for r1-Qwen-distill-1.5B

| Trace Style | GRPO | both | | | | recent | | | |
| Update Style | GRPO | ε-trace | | ε-weight | | ε-trace | | ε-weight | |
| λ | GRPO | 0.98 | 0.99 | 0.98 | 0.99 | 0.98 | 0.99 | 0.98 | 0.99 |
| --- | --- | --- | --- | --- | --- | --- | --- | --- | --- |
| Trained on math12k | | | | | | | | | |
| AIME | 0.133 | 0.033 | 0.133 | 0.100 | 0.167 | 0.133 | 0.167 | 0.133 | 0.133 |
| MATH | 0.640 | 0.656 | 0.690 | 0.712 | 0.752 | 0.708 | 0.708 | 0.632 | 0.698 |
| AMC | 0.398 | 0.410 | 0.458 | 0.446 | 0.482 | 0.410 | 0.494 | 0.325 | 0.422 |
| OLBen | 0.264 | 0.277 | 0.279 | 0.311 | 0.339 | 0.289 | 0.283 | 0.247 | 0.268 |
| MIN | 0.199 | 0.202 | 0.206 | 0.158 | 0.243 | 0.210 | 0.180 | 0.143 | 0.180 |
| AVG | *0.327* | 0.316 | 0.353 | 0.345 | **0.397** | 0.350 | 0.366 | 0.296 | 0.340 |
| Trained on math-rl-16k | | | | | | | | | |
| AIME | 0.000 | 0.133 | 0.100 | 0.067 | 0.067 | 0.133 | 0.167 | 0.000 | 0.133 |
| MATH | 0.658 | 0.564 | 0.630 | 0.714 | 0.492 | 0.694 | 0.726 | 0.396 | 0.686 |
| AMC | 0.373 | 0.434 | 0.434 | 0.446 | 0.301 | 0.422 | 0.446 | 0.253 | 0.422 |
| OLBen | 0.255 | 0.271 | 0.270 | 0.299 | 0.197 | 0.305 | 0.323 | 0.141 | 0.256 |
| MIN | 0.154 | 0.180 | 0.184 | 0.228 | 0.217 | 0.224 | 0.210 | 0.110 | 0.184 |
| AVG | *0.288* | 0.316 | 0.323 | 0.351 | 0.255 | 0.356 | **0.374** | 0.180 | 0.336 |
| Trained on gsm8k | | | | | | | | | |
| AIME | 0.200 | 0.133 | 0.067 | 0.133 | 0.133 | 0.133 | 0.100 | 0.100 | 0.167 |
| MATH | 0.756 | 0.680 | 0.628 | 0.706 | 0.726 | 0.736 | 0.704 | 0.746 | 0.742 |
| AMC | 0.494 | 0.422 | 0.373 | 0.446 | 0.482 | 0.446 | 0.410 | 0.518 | 0.506 |
| OLBen | 0.330 | 0.271 | 0.255 | 0.305 | 0.324 | 0.311 | 0.283 | 0.338 | 0.329 |
| MIN | 0.243 | 0.151 | 0.162 | 0.199 | 0.239 | 0.199 | 0.213 | 0.232 | 0.261 |
| AVG | *0.405* | 0.331 | 0.297 | 0.358 | 0.381 | 0.365 | 0.342 | 0.387 | **0.401** |
| Trained on orzmath57k | | | | | | | | | |
| AIME | 0.133 | 0.067 | 0.133 | 0.133 | 0.133 | 0.167 | 0.133 | 0.167 | 0.133 |
| MATH | 0.644 | 0.684 | 0.646 | 0.726 | 0.740 | 0.728 | 0.728 | 0.686 | 0.748 |
| AMC | 0.400 | 0.422 | 0.349 | 0.518 | 0.470 | 0.422 | 0.494 | 0.386 | 0.470 |
| OLBen | 0.262 | 0.286 | 0.292 | 0.350 | 0.354 | 0.305 | 0.329 | 0.284 | 0.311 |
| MIN | 0.169 | 0.186 | 0.195 | 0.220 | 0.254 | 0.217 | 0.239 | 0.176 | 0.191 |
| AVG | *0.321* | 0.329 | 0.323 | 0.390 | **0.390** | 0.366 | 0.385 | 0.340 | 0.371 |

Table 5: Benchmark results for R1-Distill-Llama-3B

| Trace Style | GRPO | both | recent | GRPO | both | recent |
|---|---|---|---|---|---|---|
| Update Style | GRPO | $\epsilon$-trace | $\epsilon$-trace | GRPO | $\epsilon$-trace | $\epsilon$-trace |
| $\lambda$ | GRPO | 0.99 | 0.99 | GRPO | 0.99 | 0.99 |
| Trained on | | | math12k | | | gsm8k |
| AIME | 0.100 | 0.033 | 0.067 | 0.033 | 0.033 | 0.067 |
| MATH | 0.458 | 0.460 | 0.432 | 0.272 | 0.524 | 0.518 |
| AMC | 0.193 | 0.157 | 0.205 | 0.145 | 0.277 | 0.205 |
| OLBen | 0.129 | 0.124 | 0.121 | 0.090 | 0.179 | 0.166 |
| MIN | 0.165 | 0.165 | 0.217 | 0.088 | 0.132 | 0.154 |
| AVG | *0.209* | 0.188 | **0.208** | *0.126* | **0.277** | 0.222 |
| Trained on | | | orzmath57k | | | math-rl-16k |
| AIME | 0.000 | 0.000 | 0.000 | 0.033 | 0.000 | 0.000 |
| MATH | 0.300 | 0.434 | 0.410 | 0.356 | 0.214 | 0.440 |
| AMC | 0.024 | 0.157 | 0.193 | 0.108 | 0.060 | 0.205 |
| OLBen | 0.046 | 0.135 | 0.117 | 0.105 | 0.052 | 0.173 |
| MIN | 0.096 | 0.195 | 0.151 | 0.107 | 0.121 | 0.165 |
| AVG | *0.093* | **0.184** | 0.174 | *0.142* | 0.089 | **0.197** |

## E.2 DEEPSEEK-R1-DISTILL-QWEN-7B

Table 6: Benchmark results for r1-Qwen-distill-7B

| Trace Style | GRPO | recent | GRPO | recent | GRPO | recent | GRPO | recent |
|---|---|---|---|---|---|---|---|---|
| Update Style | GRPO | $\epsilon$-trace | GRPO | $\epsilon$-trace | GRPO | $\epsilon$-trace | GRPO | $\epsilon$-trace |
| $\lambda$ | GRPO | 0.99 | GRPO | 0.99 | GRPO | 0.99 | GRPO | 0.99 |
| Trained on | | math12k | | math-rl-16k | | gsm8k | | orzmath57k |
| AIME | 0.200 | 0.200 | 0.167 | 0.267 | 0.167 | 0.233 | 0.267 | 0.167 |
| MATH | 0.778 | 0.824 | 0.774 | 0.792 | 0.770 | 0.766 | 0.778 | 0.820 |
| AMC | 0.494 | 0.566 | 0.518 | 0.578 | 0.458 | 0.458 | 0.494 | 0.470 |
| OLBen | 0.361 | 0.388 | 0.356 | 0.397 | 0.353 | 0.344 | 0.375 | 0.407 |
| MIN | 0.305 | 0.338 | 0.342 | 0.346 | 0.335 | 0.298 | 0.298 | 0.353 |
| AVG | *0.428* | **0.463** | *0.431* | **0.476** | *0.416* | **0.420** | *0.442* | **0.443** |

## E.3 GENERATION LENGTHS

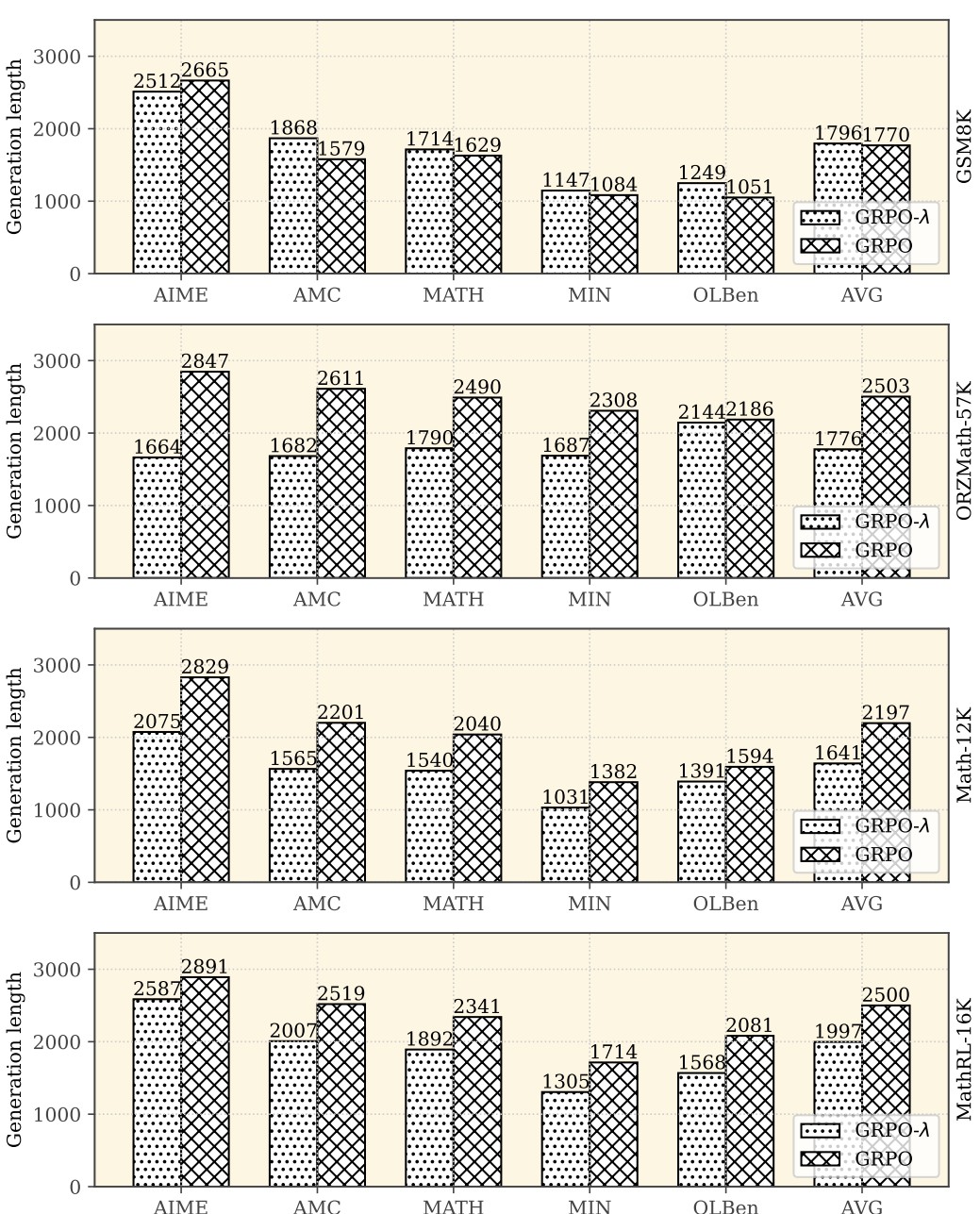

Figure 12: Comparison of the average generation length during evaluation between GRPO and GRPO-$\lambda$ when post-trained on Qwen2.5-Math-1.5B-Instruct. GRPO-$\lambda$ shows the average length across hyperparameter settings.

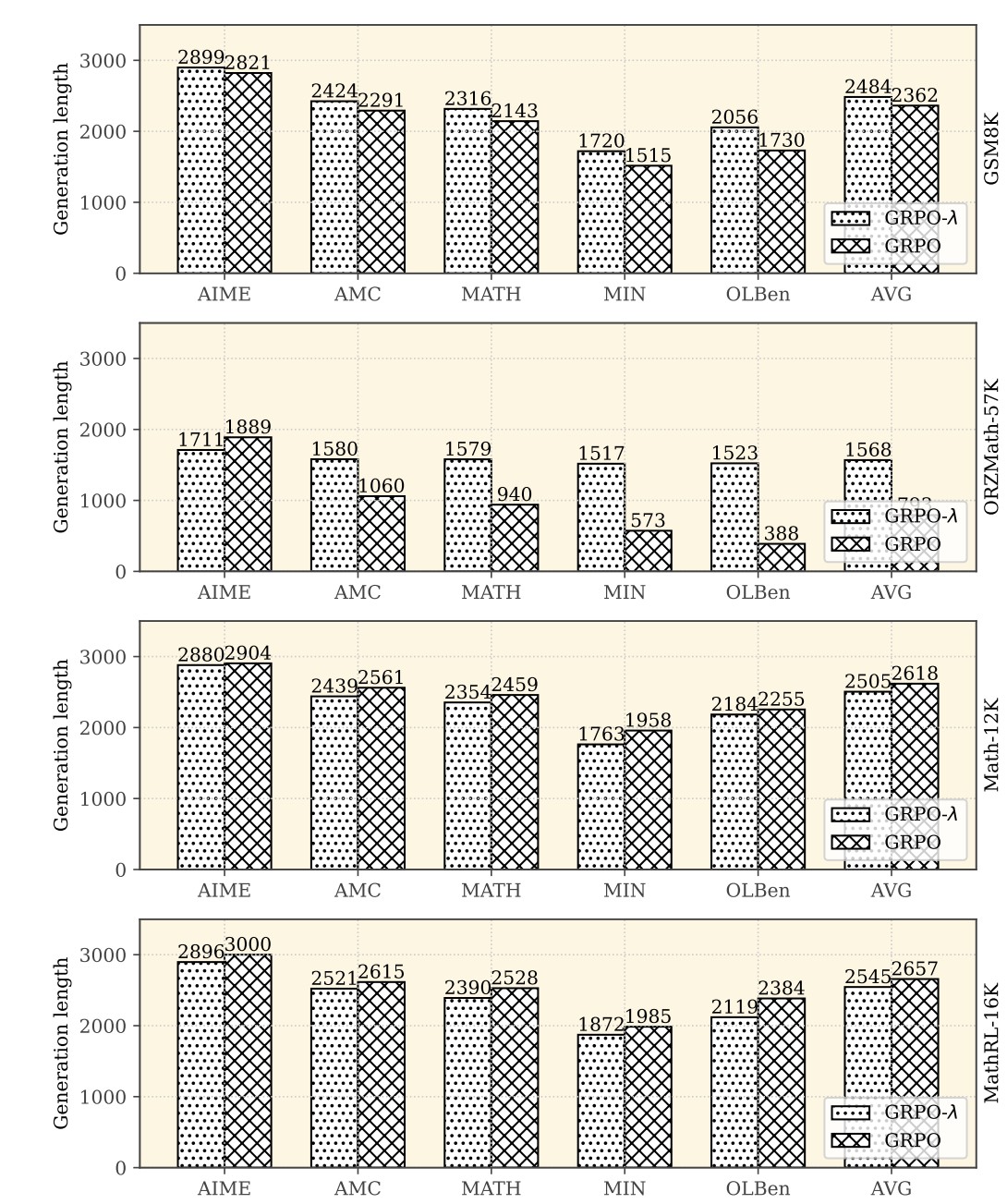

Figure 13: Comparison of the average generation length during evaluation between GRPO and GRPO-$\lambda$ when post-trained on r1-Qwen-distill-1.5B. GRPO-$\lambda$ shows the average length across hyperparameter settings.

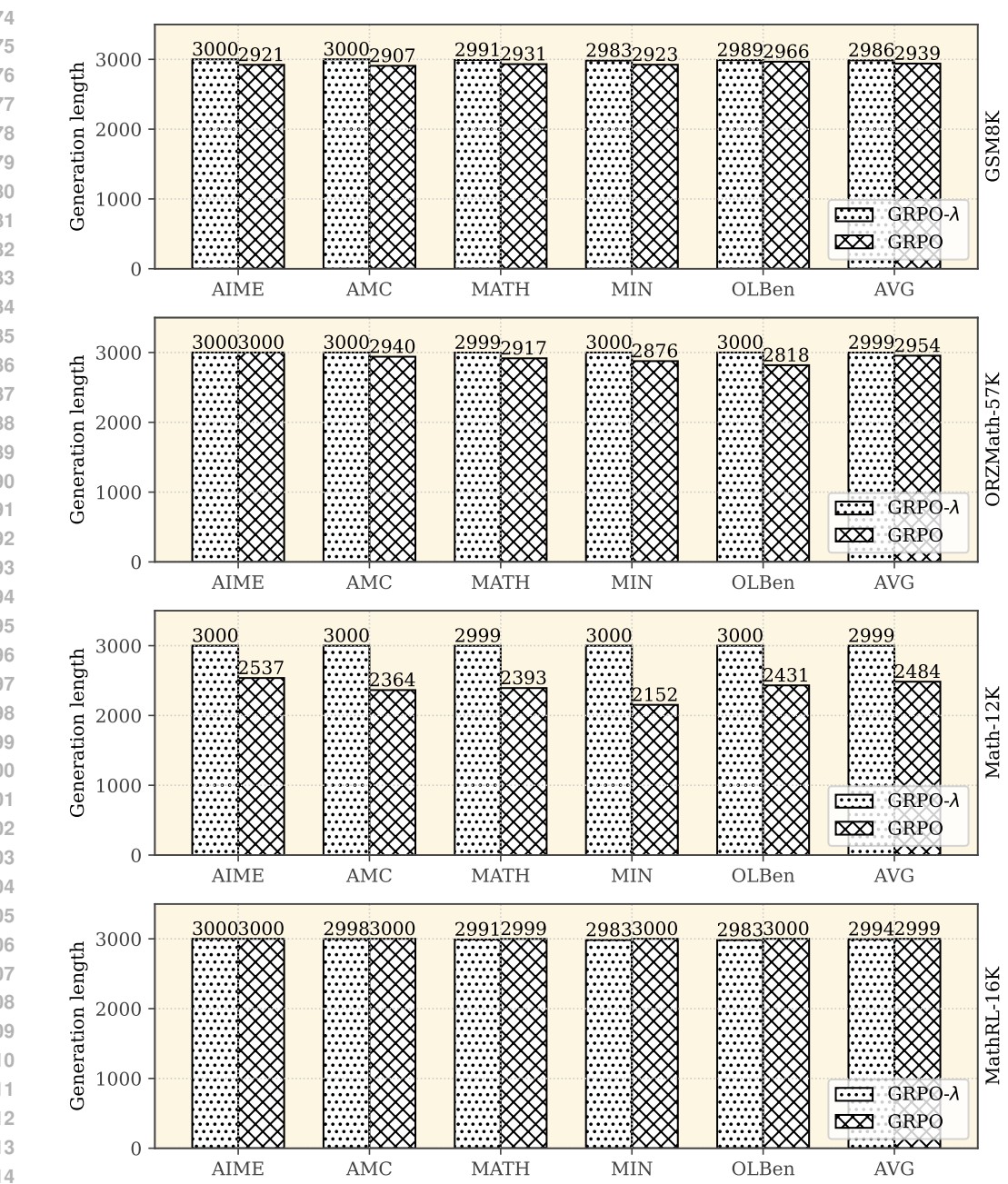

Figure 14: Comparison of the average generation length during evaluation between GRPO and GRPO-$\lambda$ when post-trained on r1-R1-Distill-Llama-3B. GRPO-$\lambda$ shows the average length across hyperparameter settings.

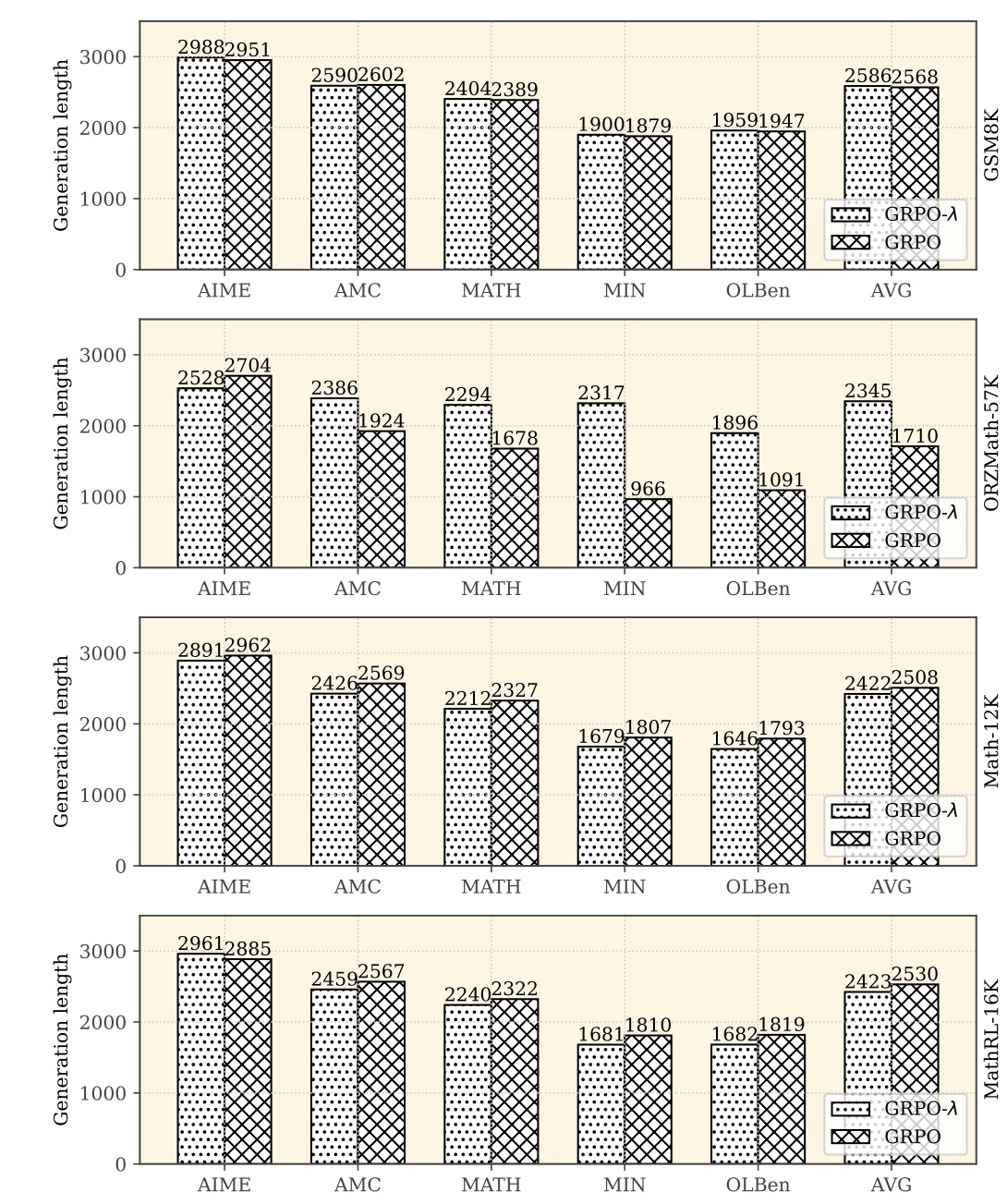

Figure 15: Comparison of the average generation length during evaluation between GRPO and GRPO-$\lambda$ when post-trained on r1-r1-Qwen-distill-7B. GRPO-$\lambda$ shows the average length across hyperparameter settings.

# F  RELATED WORK

Recent advances in the scalable training of large architectures Kaplan et al. (2020); Chowdhery et al. (2023), the development of extensive pretraining corpora Wei et al. (2022a), and refined fine-tuning strategies such as instruction tuning Zhang et al. (2023) have substantially improved the capabilities of large language models (LLMs). These improvements have enabled LLMs to produce compelling responses across a wide range of tasks, including both closed- and open-ended question answering. In parallel, significant research has been devoted to minimizing undesirable behaviors through methods broadly categorized under *preference learning* or *alignment techniques*.

## F.1  IMPROVING ALIGNMENT THROUGH PREFERENCES

**Pairwise preferences**  LLMs trained via next-token prediction often fall short in instruction-following tasks, frequently generating toxic or untruthful content. The RLHF (Reinforcement Learning with Human Feedback) framework for LLMs introduced by InstructGPT Ouyang et al. (2022) addressed this by learning a reward model $r(x, y)$ that scores responses $y$ conditioned on prompts $x$. In the pairwise setting, given a preferred response $y_w$ and a less desirable response $y_l$, the model defines the preference likelihood using the Bradley-Terry model: $P(y_w > y_l | x) = \sigma(r(x, y_w) - r(x, y_l))$ Bradley & Terry (1952).

To improve upon the original RLHF approach, DPO (Direct Preference Optimization) Rafailov et al. (2023) proposed a reparameterization of the PPO-based objective that eliminates the need for an explicit reward or value model. Subsequent variants such as $\beta$-DPO Wu et al. (2024), sDPO (Stepwise DPO) Kim et al. (2024), and TDPO (Token-level DPO) Zeng et al. (2024) aim to enhance stability, mitigate overfitting, and preserve generation diversity.

**Extensions with binary and listwise preferences**  Several efforts have explored alternative forms of preference data to reduce annotation burdens and improve learning. KTO (Kahneman-Tversky Optimization) Ethayarajh et al. (2024) and DRO (Direct Reward Optimization) Richemond et al. (2024) use binary feedback instead of pairwise comparisons, avoiding the need to collect pairwise preferences. KTO incorporates principles from prospect theory, introducing hyperparameters $\alpha$ and $\lambda$ to shape the value function's curvature and steepness. In contrast, DRO learns a parameterized value function jointly with the policy, showing superior empirical results relative to KTO. Alternativvely, LiPO (Listwise Preference Optimization) Liu et al. (2024) extends pairwise preferences by utilizing listwise preference data, arguing that richer signals from ranked outputs enable better alignment. However, the approach is sensitive to data quality and requires non-trivial filtering to remove noise from the training signal.

**Advanced CoT with on-policy samples**  LLMs are increasingly applied to complex domains such as scientific QA, mathematical reasoning, and code generation. With sophisticated pretraining and high quality SFT, Ding et al. (2023); Xu et al. (2023a;b) noted that variance in policy updates were no longer an issue. This variance reduction resulted in RLOO (REINFORCE Leave-one-out) (Ahmadian et al., 2024), that uses multiple on-policy samples to estimate the baseline for the REINFORCE policy gradient update. RLOO demonstrated significant performance improvement over DPO and PPO especially when more on-policy samples can be generated. GRPO (Shao et al., 2024), a related method, avoids the leave-one-out step by estimating normalized advantages using a $z$-score across sampled completions. DeepseekMath, Deepseek-R1, and Deepseek-R2 all utilize GRPO for their significantly superior reasoning trajectories.

**Improvements to GRPO**  GRPO originally aggregates token-level losses normalized by sequence length, which introduces a length bias favoring shorter responses. Dr. GRPO (Liu et al., 2025) mitigates this by normalizing over the maximum completion length instead. Other extensions to GRPO include BNPO (batch normalized GRPO) (von Werra et al., 2020), which introduces a minor yet effective modification: loss normalization across active tokens in a batch. When the `batch_size=1` the loss behaves like the orginal GRPO loss. DAPO Yu et al. (2025) decouples the PPO clipping parameter into $\epsilon_{high}$ and $\epsilon_{low}$, and employs dynamic resampling to maintain meaningful gradients when batch rewards are either all 0 or all 1. GRPO has also been extended with improvements such as explicit penalties for undesirable responses, length-aware reward shaping, and difficulty-weighted advantage scaling (Zhang & Zuo, 2025). Complementary to these, GRPO-$\lambda$

proposes trace-weighted advantage estimation for better credit assignment, accelerating learning and improving robustness on challenging benchmarks.

### F.2 IMPROVING LLM REASONING

**Training for improved credit assignment** RLHF (Christiano et al., 2017; Ouyang et al., 2022), based on PPO Schulman et al. (2017), relies on explicit value models for reward and baseline estimation. DPO Rafailov et al. (2023), in contrast, treats the response as a single bandit action, eliminating the need for value modeling. GRPO (Shao et al., 2024) and RLOO (Ahmadian et al., 2024) similarly avoid explicit critics, instead estimating baselines from multiple samples. While value models can accelerate learning, they can suffer from drift, causing misalignment between the critic and policy. VinePPO (Kazemnejad et al., 2024) addresses this via Monte Carlo rollouts from intermediate states, yielding more accurate value estimates. Beyond architectural modifications, recent work has explored leveraging both positive and negative samples to enhance learning. Setlur et al. (2024) show that incorporating negative trajectories helps unlearn spurious correlations and establish a connection to advantage-weighted reinforcement learning. In a similar vein, Hwang et al. (2024) propose Self-Explore, wherein the model identifies its first incorrect reasoning step and generates multiple continuations to construct step-level preference data. This enables fine-grained updates via DPO and leads to improved reasoning capabilities.

**Inference-time reasoning enhancements** In addition to their role in training, value estimates have proven effective during inference, particularly in planning-based approaches such as AlphaGo Silver et al. (2016) and AlphaZero Silver et al. (2017), which use tree search guided by value networks. Analogous strategies have been adapted for LLMs to enhance inference-time reasoning. Several approaches operate without explicit value critics but instead rely on structured prompting or model-internal heuristics. Tree-of-Thoughts prompting (Yao et al., 2023) enables models to generate multiple intermediate reasoning paths and iteratively evaluate them to choose the most promising trajectory. Alternatively, Weng et al. (2022) checks the correctness of their own intermediate outputs through self-verification to improve the quality of CoT generation, while Shinn et al. (2023) reflects over the partial generation to improve and align better with the preferences and prompt. Planning-based techniques take this further by explicitly decomposing a complex input query into a sequence of subproblems (Wang et al., 2023a). Even lightweight inference-time strategies like self-consistency decoding (Wang et al., 2022) have demonstrated performance gains, outperforming deterministic decoding strategies such as greedy or beam search.

