# OpenReview forum: "GRPO-$\lambda$: Credit Assignment improves LLM Reasoning"
_ICLR.cc/2026/Conference — ICLR 2026 Conference Withdrawn Submission_

### Official Review · Reviewer_K44U · 2025-10-28

**Soundness:** 2
**Presentation:** 3
**Contribution:** 2
**Rating:** 2
**Confidence:** 3

**Summary:**

The paper proposes GRPO-λ, a critic-free, token-level credit-assignment variant of GRPO using λ-style returns/eligibility traces. On math-reasoning tasks with multiple models (1.5B–7B), the authors report faster learning and higher final accuracy than vanilla GRPO.

**Strengths:**

1. Theory and token-weighting are well-motivated. It introduces a principled and lightweight way to introduce token-level credit assignment into a critic-free GRPO pipeline. Per-token weights are consistent with a λ-return reparameterization, mirroring eligibility-trace coefficients $(\gamma \lambda)^{l}$.
2. Implementation is simple and broadly applicable within GRPO-style RL.

**Weaknesses:**

My primary concern is the problematic experimental setting (length budget vs. model type). Three of the four backbones are R1-distilled models, but the paper does not compare against their reported baselines. In fact, all reported results are far below the official R1 numbers. For example, DeepSeek-R1-Distill-Qwen-7B [1] is reported at 55.5 / 92.8 on AIME24 / MATH-500, whereas after GRPO-λ training the paper reports only 21.7 / 80.0. This raises concerns that the experimental setup itself is suppressing performance rather than demonstrating improvement.

I found that the paper trains with too short max completion lengths for distilled models (e.g., 256 for 1.5B/3B and 768 for 7B), while it looks like that evaluation allows up to 3k tokens (Fig. 12–15), and the average generation length already saturates the 3k on several benchmarks. That implies a substantial fraction of answers may be truncated, making the reported accuracies unreliable (e.g., answers counted as wrong simply because they could not finish). The paper does not (i) clearly specify eval length, (ii) report the fraction truncated. I also checked the Qwen2.5-Math-1.5B-Instruct cases; these do not dispel the concern that accuracy is driven by less-truncation.


[1] DeepSeek-R1: Incentivizing Reasoning Capability in LLMs via Reinforcement Learning

**Questions:**

1. Can author provide experiments with substantially larger training and evaluation budgets.
2. Can you report base backbone performance to verify that your RL training actually improve the model's performance?

---

### Official Review · Reviewer_UYiH · 2025-10-31

**Soundness:** 3
**Presentation:** 3
**Contribution:** 3
**Rating:** 8
**Confidence:** 3

**Summary:**

The paper introduces GRPO-λ, a novel extension of the Group Relative Policy Optimization (GRPO) algorithm designed to enhance the reasoning abilities of Large Language Models (LLMs) through reinforcement learning. The core innovation is the integration of a critic-free reformulation of Generalized Advantage Estimation (GAE), which allows for more effective credit assignment across long token sequences using eligibility traces. This is achieved without the computational and memory overhead of a separate critic network. The authors theoretically analyze the value estimation bias present in the original GRPO algorithm (Lemma 1) and propose a reparameterization of the policy gradient that incorporates eligibility traces (Theorem 1). This new objective function, GRPO-λ, is shown to improve learning efficiency. The paper provides an extensive empirical evaluation across multiple model architectures (Qwen2.5, LLaMA-3.1) and sizes (1.5B, 3B, 7B) on a variety of challenging mathematical reasoning benchmarks. The results consistently demonstrate that GRPO-λ achieves faster convergence and significantly outperforms the standard GRPO baseline.

**Strengths:**

This paper presents a strong piece of research that makes a valuable contribution to the field of large language model reasoning.
1. Originality: The core originality of this work lies not in the invention of a completely new algorithm, but in the insightful synthesis of two powerful concepts from reinforcement learning: the critic-free, multi-rollout approach of Group Relative Policy Optimization (GRPO) and the temporal credit assignment mechanism of eligibility traces (λ-returns), traditionally used in actor-critic methods like PPO. The authors provide a novel, critic-free reparameterization of Generalized Advantage Estimation (GAE) (Theorem 1). This elegantly preserves the lightweight nature of GRPO while addressing its key limitation in fine-grained credit assignment. Furthermore, the paper demonstrates originality by analyzing the underlying biases of the baseline method (Lemma 1). This analysis directly motivates the exploration of an alternative trace weighting scheme ("both" style).
2. Quality: The quality of the research is high, supported by both theoretical rigor and a comprehensive empirical evaluation. The methodological contributions are well-grounded, with clear theorems and lemmas that formalize the proposed approach and its motivation. The experimental setup is thorough and convincing.
3. Clarity: The paper is written with outstanding clarity. The authors do an excellent job of motivating the problem, providing the necessary background on RL, PPO, and GRPO, and logically building up to their proposed solution, GRPO-λ. The core concepts are explained precisely. The structure is logical, and the arguments flow smoothly.
4. Significance: This research highlights the critical importance of effective credit assignment in RL-based fine-tuning for reasoning tasks. By demonstrating that propagating reward signals more intelligently to earlier tokens in a sequence leads to better performance, the paper opens up a promising avenue for future research.

**Weaknesses:**

While the paper is strong overall, there are several areas where further clarification and investigation could strengthen its contributions and address potential concerns.
1. Limited Scope of Reasoning Tasks: The experiments are exclusively focused on mathematical reasoning. While this is a challenging and important domain, "reasoning" in LLMs is a broader concept that also includes areas like code generation, logical deduction, and complex instruction following. It is unclear if the performance gains from GRPO-λ would generalize to other sequential reasoning tasks where the error propagation and credit assignment dynamics might differ.
2. Omission of a Key Conceptual Baseline: The paper compares GRPO-λ primarily against the standard GRPO. While this is the most direct ablation, it misses a comparison to other recent methods that also explicitly tackle the problem of poor credit assignment and biased value estimates in critic-free RL for LLMs. Specifically, VinePPO (Kazemnejad et al., 2024) is a highly relevant baseline, as it also aims to get more accurate value estimates for intermediate tokens, albeit through a different mechanism (Monte Carlo rollouts from intermediate states).
3. Under-explored Stability and Hyperparameter Sensitivity: The authors note in their limitations that GRPO-λ leads to a larger KL-divergence from the reference policy and requires "advantage clamping" to maintain stability. This is a critical practical detail that warrants more investigation. The paper uses a fixed clamping value of -0.1 but does not explore how sensitive the model's performance and stability are to this specific hyperparameter.

**Questions:**

1. On the Generalizability of the Method Beyond Mathematical Reasoning: Could you discuss the theoretical and practical implications of applying GRPO-λ to tasks like code generation? In coding, a bug might be localized, and the importance of tokens might not decay or accumulate in the same way as in a mathematical proof. How would eligibility traces handle credit assignment in such a scenario?
2. Contextualizing the Contribution with Respect to VinePPO: Do you have an intuition for which method might be more effective or efficient under different circumstances (e.g., for very long vs. short generation lengths)?
3. Understanding the "Both" Trace and Its Implications: The finding that the "both" trace style—which upweights early tokens—outperforms the traditional "recent" trace is fascinating and potentially very significant. The current explanation is still somewhat speculative. Could you provide a more detailed analysis of why this might be the case? Is the improved performance of the "both" style a direct consequence of correcting the value estimate bias for later tokens, as identified in your Lemma 1?
4. On Training Stability and the Role of Advantage Clamping: How did you arrive at the specific clamping value of -0.1? Could you share any results from ablations on this hyperparameter? It would be very helpful to see how performance and the KL-divergence from the reference policy are affected by different clamping thresholds.

---

### Official Review · Reviewer_9mws · 2025-11-01

**Soundness:** 2
**Presentation:** 1
**Contribution:** 2
**Rating:** 2
**Confidence:** 3

**Summary:**

This paper introduces a reparameterization of the generalized advantage estimation in GRPO that transforms the standard probability ratio in the GRPO/PPO objective as a $\lambda$-weighted objective. In addition to the standard form, the authors also propose a variant named "both" that also puts weights on tokens at the beginning of the sequence. Experiments taken on different LLMs show that the proposed approach could achieve better training results.

**Strengths:**

1. This paper provides a new perspective for investigating RLVR training.
2. This work provides theoretical analysis to the proposed approach.

**Weaknesses:**

1. Although theoretical results are provided, it is still unclear why the weighted objective could achieve a better performance than standard GRPO.
2. In recent practice of RLVR training, GAE is not used as a standard practice, which makes the paper less properly motivated.
3. It is unclear why the "both" mechanism is proposed  and the benefit of it over the "recent" trace.
4. The reported evaluation results on the LLMs are much lower than the official report of these LLMs, e.g. AIME24.

**Questions:**

See weakness.

---

### Official Review · Reviewer_oWWc · 2025-11-03

**Soundness:** 3
**Presentation:** 3
**Contribution:** 3
**Rating:** 4
**Confidence:** 2

**Summary:**

This paper proposes GRPO-λ, an extension of Group Relative Policy Optimization (GRPO) that incorporates eligibility traces for improved credit assignment in reinforcement learning fine-tuning of LLMs for reasoning tasks. The authors reformulate PPO's Generalized Advantage Estimation (GAE) eligibility traces to work in a critic-free setting by expressing them through token-level log-probabilities. The key contribution is Theorem 1, which reparameterizes the GAE policy gradient as weighted cumulative action log-probabilities, enabling λ-return approximation without a value function critic. The paper also derives Lemma 1 bounding the bias in GRPO's value estimates and proposes alternative token weighting schemes ("recent" vs. "both" trace styles, "ϵ-trace" vs. "ϵ-weight").

**Strengths:**

- Comprehensive experimental validation across multiple model sizes (1.5B, 3B, 7B) and architectures (Qwen2.5, LLaMA-3.1)
- Well-structured paper with clear motivation building from GRPO's limitations
- Addresses practical limitation of GRPO—poor credit assignment for earlier tokens

**Weaknesses:**

- No convergence guarantees or sample complexity analysis for GRPO-λ
- Missing theoretical justification for why λ=0.99 is optimal—selection appears purely empirical
- All experiments limited to mathematical reasoning tasks only
- No statistical significance testing; results presented without error bars or confidence intervals

**Questions:**

- Can you provide theoretical analysis for why λ=0.99 is optimal? Is there a principled way to select λ?
- How does GRPO-λ perform on non-math reasoning tasks (code, logic, commonsense)?

---

### Note · Authors · 2025-12-03

I have read and agree with the venue's withdrawal policy on behalf of myself and my co-authors.